# Annual net primary productivity of a cyanobacteria-dominated biological soil crust in the Gulf savannah, Queensland, Australia

Burkhard Büdel[1], Wendy J. Williams[2], Hans Reichenberger[1]

[1]Plant Ecology and Systematics, University of Kaiserslautern, Kaiserslautern, D-67663, Germany
[2] Arid Soil Ecosystems, Agriculture and Food Sciences, University of Queensland, Gatton, 4343, Australia

*Correspondence to*: Burkhard Büdel (buedel@bio.uni-kl.de)

**Abstract.** Biological soil crusts (biocrusts) are a common element of the Queensland (Australia) dry savannah ecosystem and are composed of cyanobacteria, algae, lichens, bryophytes, fungi and heterotrophic bacteria. Here we report how the $CO_2$ gas-exchange of the cyanobacteria-dominated biocrust type form Boodjamulla National Park in the north Queensland Gulf savannah responds to the pronounced climatic seasonality and their quality as a carbon sink using a semi-automatic cuvette system. The dominant cyanobacteria are the filamentous species *Symplocastrum purpurascens* together with *Scytonema* sp. Metabolic activity was recorded between July 1$^{st}$ 2010 and June 30$^{th}$ 2011 where $CO_2$ exchange was only evident from November 2010 until mid-April 2011, representative of 23.6% total time of the year. In November at the onset of the wet season, the first month (November) and the last month (April) of activity had pronounced respiratory loss of $CO_2$. The metabolic active period accounted for 25% of the wet season and of that period 48.6% were net photosynthesis (NP) and 51.4% dark respiration (DR). During the time of NP, net photosynthetic uptake of $CO_2$ during daylight hours was reduced by 32.6% due to water suprasaturation. In total, the biocrust fixed 229.09 mmol $CO_2$ m$^{-2}$ yr$^{-1}$, corresponding to an annual carbon gain of 2.75 g m$^{-2}$ yr$^{-1}$. Due to malfunction of the automatic cuvette system, data from September and October 2010 together with some days in November and December 2010, could not be analysed for NP and DR. Based on climatic and gas exchange data from November 2010, an estimated loss of 88 mmol $CO_2$ m$^{-2}$ was found for the two month, resulting in corrected annual rates of 143.1 mmol $CO_2$ m$^{-2}$ yr$^{-1}$, equivalent to a carbon gain of 1.7 g m$^{-2}$ yr$^{-1}$. The bulk of the net photosynthetic activity occurred above a relative humidity of 42%, indicating a suitable climatic combination of temperature, water availability and light intensity well above 200 µmol photons m$^{-2}$ s$^{-1}$ photosynthetic active radiation. The Boodjamulla biocrust exhibited high seasonal variability in $CO_2$ gas exchange pattern, clearly divided into metabolically inactive winter month and active summer month. The metabolic active period commences with a period (of up to 3 month) of carbon loss, likely due to reestablishment of the crust structure and restoration of NP prior to about a four-month period of net carbon gain. In the Gulf savannah biocrust system, seasonality over the year investigated showed that only a minority of the year is actually suitable for biocrust growth and thus a small window for potential contribution to soil organic matter.

## 1. Introduction

Biological soil crusts (named "biocrusts" throughout the text) are a consortium of heterotrophic bacteria, cyanobacteria, algae, fungi, lichens and bryophytes in different proportions with photoautotrophic organisms dominating their biomass. They cover dryland soil surfaces and can make up to 70% of a dryland ecosystem's living cover (Belnap, 1995; Belnap et al., 2016), but also occur in other climatic regions where competition with vascular plants is low (Büdel, 2001; Büdel et al., 2014). Due to the poikilohydric character of biocrust organisms, biocrusts exhibit a high resilience under extreme conditions and a remarkable adaptation to various combinations of climatic factors (e.g. Karsten et al., 2016; Sancho et al., 2016 and citations herein), thus making them excellent candidates for pioneering hostile environments on our planet. There is good evidence that cyanobacteria-dominated biocrusts have inhabited Earths soil surfaces at least 2600 million years ago

(Watanabe et al., 2000; for an overview see also Beraldi-Campesi and Retallack, 2016). Lalonde and Konhauser (2015) point to the importance of oxygenic photosynthesis of early biocrusts providing sufficient equivalents for oxidative-weathering reactions in benthic and soil environments. This certainly also points to the role of biocrusts in soil formation and soil fertility, for example by leaching carbon and nitrogen to initial soils. Consequently, there is growing interest in carbon gain
of biocrusts (Lange and Belnap, 2016) and their $CO_2$ exchange rates are considered relevant on local and global scales (e.g. Castillo-Monroy et al. 2011; Wilske et al., 2009; Elbert et al., 2012; Porada et al., 2013, 2014). Process based models as used by Porada et al. (2013; 2014) still rely on a few available datasets covering a small number of biocrust types, organisms, geographical regions, and climatic situations (see also summary in Sancho et al., 2016).

With the focus on $CO_2$ gas exchange of biocrusts over longer periods of time, a number of studies were published either
on the basis of long term measurements or modelled from single or grouped measurements. From these results, two biocrust groups can be distinguished, one where biocrusts experienced net C-uptake and the other where biocrusts experienced C-loss. Examples from the net C-uptake group include: a biocrust from the Mojave Desert gained 11.7 g C $m^{-2}$ $yr^{-1}$ (Brostoff et al., 2005), a biocrust of the northern Negev Desert, Israel had a net C-uptake of 0.7-5.1 g $m^{-2}$ $yr^{-1}$ (Wilske et al., 2008, 2009), and a biocrust from a desert region of northwest China showed a net C-uptake of 3.5 to 6.1 g C $m^{-2}$ $yr^{-1}$ (Feng et al., 2014).
Among the C losing biocrusts are those of southeast Utah determined to be typical net C-sources (Bowling et al., 2011), a biocrust from the Colorado Plateau, USA also losing $62 \pm 8$ g C $m^{-2}$ $yr^{-1}$ (Darrouzet-Nardi et al., 2015), and finally biocrusts from the Gurbantunggut Desert, north-western China exposed a C-release of $48.8 \pm 5.4$ to $50.9 \pm 3.8$ g $m^{-2}$ $yr^{-1}$ (Su et al., 2013). One can be fairly confident that persistent biocrusts must have a positive C-balance. If they did not, they would certainly disappear from the reference habitats. Yet, despite the plausibility that biocrusts must have a positive net C-balance,
it is difficult to observe net $CO_2$ uptake. There are various reasons however, two major ones are: 1) positive $CO_2$ uptake might only occur during a small part of the year and 2) it is difficult to separate the C-balance of the biocrust from C-fluxes of other organisms like microbes and roots of higher vegetation or minerals like carbonate that occur below them. Nevertheless, biocrusts are only one constituent of mature soils and it seems plausible that measurements that include soil layers other than the biocrust itself might result in $CO_2$ release because of a high percentage of heterotrophic organisms
(Bowling et al., 2011; Darrouzet-Nardi et al., 2015; Su et al., 2013), while those that restrict strictly to the biocrust layer might explain why they show net C-uptake over the year (Brostoff et al., 2005; Wilske et al., 2008, 2009; Feng et al., 2014). We believe that seasonality (biocrust wet-up and dry-down) plays an important role too for several reasons. For example do cyanobacterial colonies exposed to wet-dry cycles apparently not fully recover, in that quite a number of cells die during the dry period (e.g. Grilli-Caiola, et al., 1993; Grilli-Cailola and Billi, 2007). Another important observation considered in the
design of the present study was that the determination of $CO_2$ gas exchange of single species might not represent the biocrust. There is a strong influence on the outcome of the measurements when species are removed from the context of the biocrust, rather than studying the whole biocrust system (Colesie et al., 2016; Elbert et al., 2012), as this does not necessarily represent the ecological response of an intact biocrust (Weber et al., 2012).

Previously, it was observed that we could not resurrect the Australian Gulf savannah biocrusts photosynthetic activity in
the middle of the dry season, even after soaking them in water for more than 24 hours (Williams et al., 2014). This motivated us to perform a long term study on an entire cyanobacteria-dominated biocrust common in northern Queensland, considered a mid-successional type in a highly seasonal environment. All these considerations led us to the questions: 1) how do the cyanobacteria-dominated biocrusts of Boodjamulla respond to the pronounced seasonality of water availability and 2) are these biocrusts sources or sinks for carbon at an annual timescale?

Here we focused on a common biocrust type occurring in the Gulf Plains bioregion characterized by woodlands and extensive perennial grasslands. Our investigation site was situated in Boodjamulla National Park in the Gulf Plains dry

savannah region of north-eastern Australia, established in 1985 and since then, cattle grazing ceased. Cyanobacteria-dominated biocrusts are important drivers of ecosystem function throughout Queensland's dry savannah and especially in Boodjamulla National Park (Williams et al., 2014). There is very little rainfall during the winter dry season and the vast majority of rain commences during the summer wet-season accompanied by high ambient air temperatures (>40°C) and high soil surface temperatures (60-74°C). Heavy rains in the wet season often result in vast flooded plains and ephemeral wetlands (Williams et al., 2014).

## 2. Material and methods

### 2.1 Investigation site

Boodjamulla National Park (18.39°S, 138.62°E) is situated in the Gulf Savannah of north-eastern Australia covering an area of 2,820 km$^2$. Mean annual rainfall is 641 mm falling mostly between December and February, although it can be highly variable with up to 1121 mm falling in the wet years (www.bom. gov.au). Boodjamulla is mainly situated on sandstone, limestone, calcium carbonate or tufa formations sustaining Eucalyptus and *Melaleuca* woodlands, perennial grass floodplains, Spinifex grasslands and riparian vegetation (Fig. 1a). The biocrusts of this area are dominated by the cyanobacteria *Symplocastrum purpurascens* (Gomont ex Gomont) Anagnostidis (Fig. 2a, b, d-f), *Scytonema* sp. (Fig. 2a, b), *Symploca* sp., and *Nostoc commune* Vaucher ex Bornet et Flahault as well as other *Nostoc* species. Other organisms occurring regularly in the Boodjamulla biocrust are the hairy liverwort *Riccia crinita* Taylor, the lichens *Peltula patellata* (Bagl.) Swinscow & Krog, *Heppia lutosa* (Ach.) Nyl. and *Placidium squamulosum* (Ach.) Breuss and other small non-fertile lichen species. For a more detailed description of the locality and the biocrust see Williams and Büdel (2012) and Williams et al. (2014). We selected a site next to the national park rangers station (Fig. 1a) with luxuriant biocrust growth (Fig. 1b), and to guarantee maximum control of the monitoring setup (Figs 1c-e) kindly provided by the national park rangers. The biocrust we used here for the analysis was primarily formed by the two cyanobacteria *S. purpurascens* and *Scytonema* sp. with smaller amounts of other species including *Nostoc* sp. but did not include bryophytes or lichens.

### 2.2 Sampling and sample treatment

Samples for the determination of light-, temperature- and water content related $CO_2$ gas exchange (experimental manipulations), as well as samples for the one year monitoring were collected in the direct vicinity of the instrumental setup site (Fig. 1a-c). Great care was taken for the homogeneity of all samples. For a proper collection we removed only those top soil parts stabilized by the biocrust and that was a layer between 5 – 8 mm thick using a spatula 8 cm wide. Soil particles from underneath and not fixed to the crust by any filamentous structures were removed carefully using a soft brush and tweezers.

### 2.3 Environmental manipulations

For the analysis of the effect of the different environmental factors light-, temperature- and water content (termed environmental manipulations throughout the text) on net photosynthesis, samples were air dried (at ~40°C) in a 10 cm Petri-dish, sealed and transported to the laboratory, where they were stored frozen (-20°C) until used for the measurements. This treatment had been tested in our laboratory many times with lichens of many different geographical origins, including the tropics and resulted in high survival rates (roughly 95%) compared to storing in herbarium cabinets or boxes in the laboratory. Earlier gas exchange measurements on biocrusts, cyanobacteria, bryophytes and lichens before freezing and after thawing and re-moistening resulted in identical rates (unpublished laboratory tests). Prior to the measurements, samples were dethawed at 23°C for 12 h in an air tight box a t low light intensities (<< 50 µmol photons m$^{-2}$ s$^{-1}$) in order to avoid

decondensation. Subsequently samples passively dehydrated and were kept at 23 °C and natural day-night cycles (~150 µmol photons $m^{-2}$ $s^{-1}$) for two days. Light-, temperature- and water content and related net photosynthesis (NP) measurements were performed using three independent replicates each. $CO_2$ gas exchange measurements were conducted under controlled laboratory conditions using minicuvette systems (CMS 400 and GFS 3000, Walz Company, Effeltrich, Germany). The response of NP and dark respiration (DR) was determined independently for light, temperature and water content (WC). Samples were weighed between measurements and WC was calculated later as mm precipitation equivalent after final determination of the samples dry weight (exposed 5 days in a desiccator over silica gel at the end of the measurements). To obtain the NP response to light, fully hydrated samples (n=3) were exposed to stepwise increasing photosynthetic active radiation (PAR) from 0 to 2500 µmol photons $m^{-2}$ $s^{-1}$, near optimal temperature (32°C) and ambient $CO_2$ concentration. The light cycle (about 30 min duration) was repeated until the samples were completely dry (after 3–4 h). Light saturation was defined as the PAR at 90% of maximum NP. The temperature related NP and DR were determined at increasing temperature steps, 22, 27, 32, 37, 42, and 47°C, while light was constantly at 1500 µmol photons $m^{-2}$ $s^{-1}$ and WC was constantly at optimum (n=3). The influence of WC on NP and DR was determined at constant, nearly saturating light (1500 µmol photons $m^{-2}$ $s^{-1}$) and six different temperatures (22, 27, 32, 37, 42 and 47°C), again using three replicates. Samples were completely soaked with water and exposed in the cuvette. Then, NP and DR were measured in short time intervals (roughly 10 minutes) until the samples were almost dry and did not show any NP nor DR reactions. After each time interval, the fresh weight of the sample was determined using a balance and the corresponding WC to each data point calculated using the dry weight of the sample (see above).

In all experimental manipulations, the $CO_2$ exchange rates of the samples were related to chlorophyll *a* content. For chlorophyll determination, the samples were ground to small pieces and then extracted two times with di-methyl-sulfoxide (DMSO) at 60 °C for 90 minutes. The chlorophyll a + b content was determined and calculated according to (Ronen and Galun, 1984).

## 2.4 Field monitoring of $CO_2$ gas exchange

As there was only one semi-automatic cuvette system available, we could not replicate the measurements. To partly overcome this problem, we used several samples over the year. Samples were placed in a basket of thermoplastic resin with drainholes in the bottom to avoid standing water during rain events. The basket had a fixed size and all samples had exactly the same exposed surface of 16.5 cm² (Fig. 1 d). All samples used were tested for a comparative large NP and DR rate under the given environmental conditions for two measurements (1 hour) in the cuvette system and only those were used that had more or less identical NP and DR rates. Fourteen samples were used during field monitoring (Table 1) and exposed in a random mode. The "random" mode was determined by the ability of access (climatic conditions, days off) by one of us to the investigation site during the whole measuring period.

Water content of samples of the experimental manipulations and those of the field monitoring is always expressed as millimeter water column. As it was impossible to remove the sample after each measurement from the monitoring cuvette system to determine the fresh weight corresponding with the measured value by weighing it with a balance (measurements every half an hour, none of us could stand at the site for the whole period), thus the only method of obtaining matching values between field monitoring and controlled experiments was to express it like rainfall in millimetres water column.

Field monitoring of the biocrusts $CO_2$-gas exchange was recorded using a semi- automatic cuvette system (ACS) as described in detail by Lange (2002). Full technical details of the ACS (Walz Company, Effeltrich, Germany) are given in Lange et al. (1997). We therefore focus on some major topics of the procedure here. The whole device is composed of two major parts, first the cuvette system itself that is exposed in the natural environment of the biocrust (Fig. 1c) and secondly

the controlling and data acquisition unit together with two infrared gas analysers (IRGA) for $CO_2$ ambient and $CO_2$ samples (Binos, Rosemount, Hanau, Germany) and a pumping unit regulated by mass flow controllers (Fig. 1e). For safety of securing data records a data printer and a graphics plotter were added as well.

The soil crust samples were exposed on the lower part of the cuvette (Fig. 1d, arrow). When the upper lid was open (H in Fig. 1d), the sample was fully exposed to the natural environment. Measurements were taken ==for more than one year== every 30 minutes during which the cuvette was closed for 3 min. ==Here report only the period from July 1$^{st}$ 2010 until June 30$^{th}$ 2011.== We recorded the $CO_2$ exchange of the sample and absolute ambient $CO_2$ partial pressure as well as mass flow, air temperature, the sample surface temperature, air humidity, and ambient photosynthetic radiation at the samples level. NP and DR were related to the area covered by the biocrust.

## 3. Results

The dominating cyanobacteria of the biocrust used for the long term monitoring was the felt- to tuft-like filamentous cyanobacterium *Symplocastrum purpurascens* forming a dark brownish stratum with erect tapering bunches of filaments (Fig. 2a-d) and the felt-like greyish *Scytonema* sp. inside and on top of the substratum (Fig. 2a-b). *S. purpurascens* is characterized by distinctly lamellate, reddish to purple-red sheath (colourless in shade; Fig. 2e, f). After the first rains, new trichomes developed at the tips of the cyanobacterial layer.

### 3.1 Environmental manipulations

When exposed to stepwise increasing PAR intensities, the biocrust did not reach full saturation of NP at optimal water content ($31.7 \pm 2.6$ nmol $CO_2$ mg$^{-1}$ chlorophyll *a* s$^{-1}$ at a WC of $0.7 \pm 0.1$ mm and 32°C; n= 3) even at 2500 µmol photons m$^{-2}$ s$^{-1}$. At WC below the optimal WC (i.e. WC, where at least 90% of the maximum gas exchange rates are reached), a decline of NP ($21.3 \pm 5.7$ nmol $CO_2$ mg$^{-1}$ chlorophyll *a* s$^{-1}$ at a WC of $0.5 \pm 0.1$ mm was observed. This was also the case for WC well above optimal WC, $20.7 \pm 6.2$ nmol $CO_2$ mg$^{-1}$ chlorophyll *a* s$^{-1}$ at a WC of $1.0 \pm 0.1$ mm or $7.6 \pm 3.7$ nmol $CO_2$ mg$^{-1}$ chlorophyll *a* s$^{-1}$ at a WC of $1.3 \pm 1.6$ mm and $2.3 \pm 0.2$ nmol $CO_2$ mg$^{-1}$ chlorophyll *a* s$^{-1}$ at a WC of $1.9 \pm 0.1$ mm (Fig. 3a).

Increasing air temperature from 22 to 47°C resulted in an increase of NP from $19.8 \pm 1.4$ nmol $CO_2$ mg$^{-1}$ chlorophyll *a* s$^{-1}$ to $32.4 \pm 4.5$ nmol $CO_2$ mg$^{-1}$ chlorophyll *a* s$^{-1}$ (n = 3) without saturation. The increase of DR was less expressed and ranged from -3.1 nmol $CO_2$ mg$^{-1}$ chlorophyll *a* s$^{-1}$ at 22°C to $-6.3 \pm 1.4$ nmol $CO_2$ mg$^{-1}$ chlorophyll *a* s$^{-1}$ at 47°C air temperature (n = 3; Fig. 3b).

Regarding $CO_2$ fixation, the optimal WC of the biocrust was $0.7 \pm 0.1$ mm WC. At all temperatures the biocrust exhibited a clear optimum WC (range of 0.6 to 0.8 mm) where they reached their maximum NP. Water content below and above this optimum led to a strong decline or even a complete stop of NP (Fig. 3c). At WC of about 0.2 mm the biocrust starts NP and DR and with increasing WC, NP had a steep incline to the maximum. A further increase of the WC created suprasaturation. NP then strongly decreased to less than a tenth of the maximal NP at optimal WC and could drop down to zero or even become negative at higher temperatures and remained at this level (Fig. 3c).

### 3.2 Field monitoring of $CO_2$ gas exchange

Monitoring of diurnal $CO_2$ gas exchange of Boodjamulla biocrusts commenced on July 1$^{st}$ 2010 and lasted until June 30$^{th}$ 2011. Measurements were taken every 30 minutes day and night. There was no measurable gas exchange from July until the end of September 2010 and from mid-April to June 2011. With the onset of the first seasonal rains in November, the biocrust showed mainly $CO_2$ loss during the days, despite the fact of PAR levels of 2000 - 2500 µmol photons m$^{-2}$ s$^{-1}$ (Figs. 4, 5). $CO_2$ loss during the day was in the range of up to 1 µmol m$^{-2}$ s$^{-1}$. Air temperature reached values of up to 46°C and

relative air humidity increased up to 100% during the night, dropping down to levels of 20% during the day (Fig. S1). The first positive NP was observed on November 16th (Fig. S1). From December 2010 to March 2011 rain events below 1.5 mm resulted in negative NP whereas higher precipitation initiated positive NP of up to 8 $\mu$mol $CO_2$ m$^{-2}$ s$^{-1}$ (Figs. S2-S5). In April the rainy season ceased and small precipitation events resulted in a $CO_2$ loss during the day of up to 2 $\mu$mol m$^{-2}$ s$^{-1}$ (Fig. S6). The $CO_2$ content of the ambient air fluctuated between day and night from 370 to 470 ppm during the rainy season. Fluctuation was less expressed in the dry season. During dark cloudy days with or without rain, fluctuation was diminished (Figs. S1-S6). In September and October 2010, the monitoring plot received the first rains. Nevertheless, although we could record metabolic activity, we were not able to calculate NP and DR due to malfunction of the ACS during these initial two months as well as the following days: November 1st – 2nd, 10th – 14th, 18th -20th, December 1st- 2nd, 12th, 30th – 31st and March 22nd – 23rd. These data were excluded from further analyses (see supplementary figures S1-S6). An estimation on the basis of climatic data from September and October 2010 together with gas exchange data from November 2010 resulted in an estimated $CO_2$ loss of 88 mmol m$^{-2}$.

The sensitive interaction of the biocrust and the environmental factors can be observed in the reaction of diurnal $CO_2$ gas exchange over the months. For example in the night from December 31st, 2010 to January 1st, 2011, the biocrust was inactive but did show some DR at the end of the night and positive NP was measured from the morning until the afternoon (Fig. 5). As we did not record any rain, the biocrust must have been activated by dew fall or probably from some moisture in the soil as the soil was wet the day before. In the afternoon of January 1st, a heavy rainfall occurred resulting in a strong water suprasaturation of the biocrust. Net photosynthesis immediately became negative but recovered in the late afternoon, a pattern that could also be observed on the 2nd and 3rd of January. On the 3rd of January PAR was so intensive that the biocrust dried and metabolic activity ceased completely. The biocrust did not dry on the two days before and there was DR during the whole night (Fig. 5). When comparing all positive NP values of the metabolic active period to the referring PAR and temperature values, the light saturation of the biocrust NP was reached at 2200 $\mu$mol/m$^2$ $\cdot$ s (Fig. 6a), whereas the temperature optimum was found at 37°C (Fig. 6b). The comparison of NP with relative air humidity and PAR showed that almost all of NP was found at a relative air humidity above 42% (Fig. 7).

While November 2010 and April 2011 had a negative $CO_2$/carbon balance, December 2010 to March 2011 were positive (Table 2). Net primary productivity of the Boodjamulla biocrust was 229.1 mmol $CO_2$ m$^{-2}$ yr$^{-1}$, signifying a carbon fixation rate of 2.8 g m$^{-2}$ yr$^{-1}$ (Table 2; Fig. 4). Applying the September-October estimation, annual values were reduced to 143.1 mmol $CO_2$ m$^{-2}$ yr$^{-1}$ equivalent to 1.7 g C m$^{-2}$ yr$^{-1}$. Over the 8,760 hours of the one-year measurement period, the biocrust was metabolically active for 2186 hours, representing 25% of the whole period. Of that 25% total active period, 48.6% were NP and 51.4% DR (Fig. 8a). The biocrust suffered from a reduced $CO_2$ uptake during NP periods due to water suprasaturation for over 29.2% of the photosynthetically active time (Fig. 8b).

## 4. Discussion

### 4.1 Seasonality and $CO_2$ balances

Apart from a clear seasonal activity pattern of the cyanobacteria-dominated biocrust from Boodjamulla National Park, Queensland, only a minority of the year was actually suitable for its growth during the one year round $CO_2$ gas exchange field monitoring. An inactive winter period with no measurable $CO_2$ gas exchange lasted from July to mid-September 2010 and then from mid-April to end of June 2011. Metabolic activity was found in the summer months only, starting with September 23rd 2010 where the first rains commenced, continuing until April 18th 2011. Due to malfunction of the ACS, measurements from September and October and some days of November and December 2010 were not useable to calculate NP and DR. An estimation based on rainfall data from September and October, together with the reference gas exchange

values from November suggests a $CO_2$ loss of roughly 88 mmol m$^{-2}$. Net primary productivity was determined as 1.7 g C m$^{-2}$ yr$^{-1}$ (2.8 g C m$^{-2}$ yr$^{-1}$ without Sept.-Oct. correction). Our results showed that the Boodjamulla biocrust exhibited a positive net C-uptake after one year field monitoring. This result is in line with the findings of several other studies but differs from all of them in the fact that our study focused to an environment with hot wet-season hydration, whereas all of the other studies were conducted in environments with cool season hydration. For example, a cyanobacteria-dominated biocrust in the Mojave Desert, USA had a C gain of 11.5 g m$^{-2}$ yr$^{-1}$ (Brostoff et al., 2005), 6.7 times higher than the cyanobacteria-dominated Boodjamulla biocrust. Another biocrust dominated by cyanobacteria, algae, lichens and mosses from the Negev Desert, Israel resulted in a C gain of 0.7 to 5.1 g m$^{-2}$ yr$^{-1}$ (Wilske et al., 2008, 2009) and thus is pretty close to what we observed in our study, which also corresponds with the results from biocrusts composed of cyanobacteria, lichens and mosses of the Mu Us Desert in China with a C gain of 3.5 to 6.1 g m$^{-2}$ yr$^{-1}$ (Feng et al., 2014).

On the other hand, there are several studies that clearly demonstrate that biocrusts loose C to the atmosphere. When studying a cyanobacteria-dominated biocrust of the arid grassland in southeast Utah, USA applying the Eddy covariance method, Bowling et al. (2010) could not decide if this biocrust was a sink or a source as there were some grasses involved in the plot and hence their root respiratory $CO_2$ loss influenced the $CO_2$. When these authors applied a top soil chamber for gas exchange measurements, they found the same biocrust a typical C source (Bowling et al. 2011). But still, this does not necessarily mean that overall they are a C-source. A cyanolichen-dominated biocrust from the Gurbantungut Desert, China was reported as quite a large C source with a loss of -48.8 ± 5.4 to -50.9 ± 3.8 g C m$^{-2}$ yr$^{-1}$ (Su et al., 2012, 2013) and a very similar biocrust type of the arid grassland of the Colorado Plateau, USA that exposed surprisingly similar values of -62 ± 8 g C m$^{-2}$ yr$^{-1}$ (Darrouzet-Nardi et al., 2015). How can this astonishing and at first glance contradictory fact be explained? Comparing methodology and how measurements were taken, sheds some light on this phenomenon. All investigations, including our own study that showed biocrusts having a net $CO_2$-uptake over the year used gas exchange devices with a separate cuvette where the samples had to be removed from the biocrust (Brostoff et al., 2005; Feng et al., 2014) except the study of Wilske et al. (2008, 2009) that used a top soil chamber measuring the biocrust in situ. All other studies used top soil chambers where the biocrust is measured in situ (Bowling et al., 2011; Su et al., 2013; Darrouzet-Nardi et al., 2015). The main difference we could find in all of these studies was the thickness of the biocrust plus sub-crust (soil) layer used. While those studies revealing biocrusts as $CO_2$ losers used collars penetrating 20 to 35 cm deep into the soil (Bowling et al., 2011; Su et al., 2013; Darrouzet-Nardi et al., 2015), the studies attributing biocrusts as $CO_2$ winners during the course of one year, either used pieces of biocrusts from 1 to 5 cm thickness (this study, Brostoff et al., 2005; Feng et al., 2014), or a collar penetrating only 5.5 cm into the soil (Wilske et al., 2008, 2009). The metabolic activity of heterotrophic organisms as well as respiration of roots from nearby plants of deeper soil levels apparently influence the $CO_2$ gas exchange measurements accordingly as was already indicated in the investigation of Bowling et al. (2011). Yet, soils are not a perpetual motion machine in terms of carbon balance, they can only respire as much carbon as introduced into the system. If carbon does not come from the autotrophic part of the soil system, it must be introduced from outside, either via litter transport, blown dust, animals, or with run-on water from the surrounding environment. In a recent study using the Eddy covariance method, Biederman et al. (2017) found a wide range of carbon sink/source function. There was a mean annual net ecosystem productivity (NEP) varying from -350 to +330 g C m$^{-2}$ across sites with diverse vegetation types in the dryland ecosystems of southwestern North America using evapotranspiration (ET) as a proxy for annual ecosystem water availability. Gross ecosystem productivity (GEP) and ecosystem respiration ($R_{eco}$) were negatively related to temperature, both interannually within sites and spatially across sites and sites demonstrated a coherent response of GEP and NEP to anomalies in annual ET. Their investigation sites included one region having a noteworthy biocrust cover not accompanied by a dense vascular plant vegetation, the La Paz region of Baja California with an annual C-uptake (NEP) of roughly 90 g m$^{-2}$. Approximating annual C gain based on the maximal $CO_2$ uptake rates of four biocrust types composed of cyanobacteria, cyanolichens and

chlorolichens from Baja California (Büdel et al., 2013), we approach an annual C gain of those biocrusts of $11 \pm 4$ g m$^{-2}$. The calculation was based on the following estimations: 90 active days per year with 34 of them having a sub-optimal $CO_2$ uptake rate of only 25% of maximum due to suprasaturation. Daily rates were calculated by maximum NP for 5 hours per day minus 10 hours R + DR. This final estimation resulted in 6.5 times higher C-uptake than our pure cyanobacteria-dominated biocrust from Boodjamulla but is still 8 time less than found for the Baja California site in the study of Biederman et al. (2017). It could well be that later successional biocrusts with a wealth of different species groups, including bryophytes, lichens and green algae besides of cyanobacteria might reach a higher productivity and thus higher annual carbon fixation rates. This should be in the focus of further studies.

## 4.2 Carbon dioxide uptake rates and biocrust type

Maximum net $CO_2$ uptake rates of the Boodjamulla biocrust (8.3 µmol $CO_2$ m$^{-2}$ s$^{-1}$) clearly exceeded those of a comparable cyanobacteria-dominated biocrust from the Negev Desert, Israel that reached maximal values of 1.1 µmol $CO_2$ m$^{-2}$ s$^{-1}$ (Lange et al., 1992) and from the Colorado Plateau, USA with 2.0 µmol $CO_2$ m$^{-2}$ s$^{-1}$ (Darrouzet-Nardi et al., 2015). The higher NP rates of the Boodjamulla biocrusts are probably related to the felt like structure on the soil surface offering a higher surface for gas exchange (Fig. 2a-d), while the Negev Desert biocrust was a thin layer of cyanobacterial filaments slightly beneath the surface. Annual carbon fixation rates of cyanobacteria-dominated arid region biocrusts are generally lower (this study; Brostoff et al., 2005; Feng et al., 2014; Wilske et al., 2009) compared with biocrusts including lichens and bryophytes (see summarizing table 15.2 in Sancho et al., 2016; Elbert et al., 2012; Porada et al., 2013) or the carbon gain of isolated biocrust organisms, for example the green algal lichen *Lecanora muralis* (Schreber) Rabenh. from a rock crust with 21.5 g C m$^{-2}$ yr$^{-1}$ (Lange 2002, 2003a, b). It follows that the developmental stage of a biocrust might be an important factor for photosynthetic performance too. This view is supported by the results of a study determining NP rates depending on successional stages rather than developmental stages. In this study the authors classified biocrusts as early successional (*Microcoleus*) or later successional stages (*Nostoc/Scytonema* or *Placidium/Collema*) and differences in NP, which was on average 1.2-2.8-fold higher in later successional crusts compared to the early successional stages (Housman et al., 2006). Considering the Boodjamulla biocrust as a mid-successional type, where an increase in carbon gain might be expected in the future when lichens and bryophytes establish to form a later successional soil crust. Dealing here with a cyanobacteria-dominated biocrust of a mid-successional type might explain the low C-uptake rates and also the seeming discrepancy in the values calculated for the global NPP by cryptogamic covers of Elbert et al. (2012).

## 4.3 Active times and water relation

The Boodjamulla biocrust had metabolic activity for only 25% of the year, made up of 12.3% NP and 12.8% DR (Fig. 8a). In 29.2% of the photosynthetic active time $CO_2$ fixation was considerably lowered by water suprasaturation. For comparison, the lichen *L. muralis* from a temperate climate was active for 35.5% of year, made up of 16.7% NP and 18.9% DR. During periods of photosynthesis, the lichen was heavily depressed by water suprasaturation at 38.5% (Lange, 2003a, b). It is obvious that the strict seasonal rainfall pattern is a major contributing reason for the considerably lower metabolic activity of the savannah-type biocrust from Boodjamulla compared to the rock crust lichen *L. muralis* in a temperate climate with rainfall expanding over the whole year. As characteristic for poikilohydric organisms, both the Boodjamulla biocrust as well as the rock crust lichen suffer considerably from water suprasaturation causing waterlogged gas diffusion channels and thus drastically limiting $CO_2$ gas exchange (see Green et al., 2011 and references therein).

As metabolic activity is strictly bound to the presence of water it is important to know the role of water content on photosynthetic performance and respiratory $CO_2$ exchange. The Boodjamulla biocrust achieved maximum NP values at 0.5-0.8 mm WC and had a lower compensation point for NP at 0.1 mm WC. Comparable values were found for the Negev

biocrust studied by Lange et al. (1992) and the rock crust lichen *L. muralis* (Lange, 2002). In the chlorolichens of a biocrust from Utah, *Diploschistes diacapsis* (Ach.) Lumbsch, *Psora cerebriformis* W. Weber, and *Squamarina lentigera* (Weber) Poelt photosynthetic metabolism was activated by extremely small amounts of moisture. The lower compensation point for NP is between 0.05 and 0.27 mm WC. Maximal NP occurred between 0.4 and 1.0 mm WC (Lange et al., 1997). The values for the cyanobacterial soil crust lichen *Collema tenax* (Swartz) Ach. however, were considerably higher with the lower NP compensation point at 0.2 mm WC and maximal NP between 0.8 to 1.2 mm WC, but performed NP under much higher temperatures than the above mentioned green algal lichens (Lange et al., 1998).

Almost all gas exchange activity of the Boodjamulla biocrusts occurred at air relative humidity above 42% (Fig. 7). This however, must be taken with care as it does not mean that the biocrust is active at this value and above. Like all cyanobacteria investigated so far, the cyanobacteria of the Boodjamulla biocrusts are also not activated by air humidity alone (e.g. Lange et al., 1992, 1993, 1994 and own unpublished results). The value of 42% relative humidity is merely a good indicator for the right combination of WC (rainfall dependent), temperature and light. A comparable observation has been made by Raggio et al. (2017), who found air relative humidity ($<< 50\%$) and air temperature as the best predictors of metabolic activity duration for four different biocrust types across Western Europe. In a number of cases we found activation of the Boodjamulla biocrust without any measurable precipitation (Fig. 5, supplementary figure S3, January 4[th]-5[th]). This is likely explained by dew formation, a non-rainfall water source found playing an important role in biocrusts (Lange et al., 1994; Ouyang and Hu, 2017) and also observed at Boodjamulla during the wet season. There are a number of studies that found dew formation was important in biocrust systems, for example the study of Jacobs et al. (2000), where in a desert environment of Israel daily amounts of dew ranged between 0.1 mm/night and 0.3 mm/night. Dew formation determined for an inland dune biocrust community in Germany ranged from 0.04 kg/m² and 0.18 kg/m² within 2 days (Fischer et al., 2012). Even fog was identified as a major source of non-rainfall water driving biocrust productivity in the Atacama Desert of Chile, where approximately 8% to 24% of the fog water flux available to the biocrusts at the soil surface (Lehnert et al., 2017).

### 4.4 Reestablishment and resurrection after the dry season

What are the reasons for negative C-balances of the biocrusts during the first active months after start of the rainy season? We suggest that in contrast to eukaryotic poikilohydric photoautotrophs such as liverworts, mosses or lichens, that resuscitate all thallus compartments after hydration, prokaryotic cyanobacteria show considerable die back rates during longer dry periods or drought events (see Williams and Eldridge, 2011; Williams and Büdel, 2012). For example in the terrestrial, colony-forming unicellular genus *Chroococcidiopsis* the number of viable cells decreased with age of the colony and the length of exposure to drought (Grilli-Caiola et al. 1993; Grilli-Caiola and Billi, 2007). Desiccation-tolerant *Chroococcidiopsis* cells must either protect their components from desiccation-induced damage or repair it after rehydration. It was found that desiccation survivors limit genome fragmentation, preserve intact plasma membranes, and have spatially reduced reactive oxygen species accumulation and dehydrogenase activity whereas damaged cells do not (Billi, 2009). In the abundant biocrust cyanobacterium *Microcoleus vaginatus* Gomont ex Gomont immediate, but transient induction of DNA repair and regulatory genes signalled the hydration event and recovery of photosynthesis occurred within 1 hour accompanied by upregulation of anabolic pathways (Rajeev et al., 2013). In general, during the desiccated period homoiochlorophyllous cyanobacteria (maintain their chlorophyll during desiccation) still suffer from photoinhibition induced by the typical high light intensities of their habitat. Nevertheless, resurrection of photosynthesis after desiccation occurs within hours or days, depending on the degree of damage (Lüttge, 2011), while reestablisment takes days or weeks, largely depending on the availability of water, but also needs a positive C input. Our field monitoring uncovered numerous events of suprasaturation during daylight combined with low NP rates after the onset of the active season. Suprasaturation events later in the season are easily compensated by high NP rates (Fig. 4). We interpret the early C-loss phase after the

drought, at least partly as a reestablishment period of the biocrusts structure, enabling the biocrust diminishing suprasaturation events by means of erect cyanobacterial filament bundles, standing out of a covering water film and thus probably improve $CO_2$ gas diffusion (Fig. 2c, d).

**4.5 Influence of temperature and global warming**

In the experimental manipulation setup the Boodjamulla biocrusts did respond to air higher air temperatures (20 - 47°C) with a continuous increase of both, NP and DR, with NP increasing at slightly higher rates than DR (Fig. 3b). Even at 47°C, $CO_2$ uptake did not show any reduction nor did DR show a considerably stronger $CO_2$ release and NP still exceeded DR five times. However, in our field measurements we did not observe biocrust activity above air temperatures of 43°C (Fig. 6b), at this temperatures the biocrust was dry and inactive. During field monitoring the optimal temperature for positive NP was around 35°C (Fig. 6b). Applying an experimental air temperature increase of 2–3°C, Maestre et al. (2013) observed a drastic reduction in a lichen-dominated biocrust cover of ca. 44% in 4 years in a dryland ecosystem in Spain. Soil $CO_2$ efflux was increased and soil net $CO_2$ uptake was reduced with the additional warming. According to the field monitoring gas exchange rates of the Boodjamulla biocrusts, we would expect even shorter activity periods under the scenario of global warming and related to that, probably lower C-uptake or even C-loss resulting in a pronounced reduction of biocrust coverage. Another indirect effect of warming could be expected when it influences rainfall amount and regime. It could be speculated that less, but heavier rain events would certainly effect the Boodjamulla biocrust by increasing suprasaturation periods resulting in lower or even no carbon gain probably also causing a pronounced reduction in coverage.

**5. Conclusion**

The Boodjamulla biocrust showed a highly seasonal photosynthesis-related metabolic activity divided into four major periods: 1) a metabolically inactive winter time; 2) onset of the photosynthetic active period, starting with roughly three month of reestablishment,  and limited  $CO_2$-uptake due to suprasaturation, and a hypothesized increased activity of heterotrophic organisms decomposing organic matter from old biocrusts ; 3) a four-month period of net C-uptake; and 4) about one month with C-loss until a complete cease of activity. During the four periods, NP and NPP rates vary strongly and thus seasonality plays an important role. It is absolutely crucial in which period of the year biocrust material is sampled for eco-physiological experiments. The cyanobacteria-dominated Boodjamulla biocrust turned out to be a small but consistent sink of carbon as it grows and potentially also contributes to the soil organic matter. From the magnitude of values it is clear that the observed C fluxes are not at all close to what a plant community can do. Methodological approaches analysing the carbon cycling of biocrusts need to critically reflect, that including or excluding sub-biocrust partitions might influence the status of the biocrust as being either considered as a sink or a source. There is an urgent need for more long-term measurements on different bio crust types and developmental stages from all climatic regions of the world.

**Acknowledgements**

We acknowledge the Waayni people, traditional owners of Boodjamulla National Park and thank the staff of Boodjamulla and Adels Grove. Special thanks to Tres McKenzie for onsite assistance. We also thank AgForce Qld and Century Mine for their financial and in-kind support. BB gratefully acknowledges financial support by the German Research Foundation (DFG).

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

**Legend to the figures**

**Figure 1:** Boodjamulla National Park, measuring site. a) Housing area for the NP-rangers and NP-administration with locality of biological soil crusts and the measuring site (white circle). b) Dark patches of a biological soil crust between grass tussocks (red scale 2.5 cm). c) Klapp cuvette system installed in two water filled basins to avoid small animals occupying the device. d) Measuring head of the clap cuvette system, the lid is open exposing the wire mesh basket with the sample (IR = infrared thermocouple, L = light sensor for PAR, T = tubing for gas exchange, VP = vibration plate ensuring a regular movement of the air when the cuvette is closed, H = light translucent head closing every 30 minutes for 2.5 minutes measurement). e) Hut with the data recording devices and control module.

**Figure 2:** a) Close-up of the dry Boodjamulla biocrust at the measuring site, grey areas are dominated by the cyanobacterium *Scytonema* sp., dark-brownish areas dominated by *Symplocastrum purpurascens*. b) Same as in a, but after rehydration. c) Cross fraction of the *S. purpurascens* dominated biocrust and its stratification (EPS = extracelluar polysaccharide sheath; LT-SEM). d) In situ top view of the *S. purpurascens* biocrust. e, f) Filament with red sheath from the top of the biocrust and from beneath with a more or less colourless sheath (f).

**Figure 3**: Response of net photosynthesis and dark respiration to water content, different PAR and temperature of the *S. pupurascens* dominated biocrust; A) response of net photosynthesis to increasing PAR at different water content at 32°C, one out of three replicates shown; B) response of net photosynthesis and dark respiration to increasing air temperature at 1500 µmol photons $m^{-2} \cdot s^{-1}$ and optimal WC, mean values of n = 3; C) response of net photosynthesis and dark respiration of three replicates to increasing biocrust water content at 1500 µmol photons/m² · s and an air temperature of 47°C.

**Figure 4**: Daily $CO_2$ balance of the Boodjamulla biocrust. Black dots = dark respiration, open circles = gas exchange during daylight. Negative values during daylight indicate either suprasaturation or water shortage.

**Figure 5**: Detail of diurnal $CO_2$ gas exchange from January 2011, showing rain events resulting in water suprasaturation of the biocrust. Blue bars indicate the approximate duration of rainfall. Green lines indicate gas exchange during daylight and black lines during the night.

**Figure 6**: a) Net photosynthesis from all days related to light intensity (PAR). The biocrust shows a saturation at 2200 µmol photon/m² •s and a slight depression at 2400 and more µmol photons/m² · s. b) Net photosynthesis from all days related to air temperature. The optimum temperature is at 35 °C but the biocrust still performs very well at 42 °C.

**Figure 7**: Contour plot of net photosynthesis of the Boodjamulla biocrust based on linear interpolation between measured values. Shown is the active period from November 2010 to April 2011. Net photosynthesis is related to relative air humidity and photosynthetic active radiation (PAR). No dark respiration values shown! Colour key: yellow = no activity, orange to red = $CO_2$ loss during the day (suprasaturation), light green to violet = $CO_2$ uptake.

**Figure 8**: A) Mean diel activity of the Boodjamulla biocrust; black = inactive, light grey = photosynthetically active, dark grey = dark respiration, hatched = metabolic activity but due to technical failure of instrumentation, not clear if NP or DR. B) Monthly extent of water suprasaturated periods during the photosynthetic (NP) active time of the Boodjamulla biocrust. Black = periods of suprasaturation, light grey = periods of conducive water supply.

**Supplementary figures**

**Figure S1-S6**: Each month with metabolic activity is shown. One month is represented as a set of three graph pairs (except April where metabolic activity ceased by mid-month), each pair composed of an upper graph showing CO2 gas exchange (green and black curves) and PAR (brown curve) and a lower graph showing ambient CO2 concentration (black curve), air temperature (red curve), and relative air humidity (blue curve).

**Figure S1**: Diel carbon dioxide gas exchange in November 2010

**Figure S2**: Diel carbon dioxide gas exchange in December 2010

**Figure S3**: Diel carbon dioxide gas exchange in January 2011

**Figure S4**: Diel carbon dioxide gas exchange in February 2011

**Figure S5**: Diel carbon dioxide gas exchange in March 2011

**Figure S6**: Diel carbon dioxide gas exchange in April 2011

**Table 1**: Samples used for monitoring (only those listed used during the active period).

        1. Sample A10             Jul. $1^{st}$ – Dec. $12^{th}$ 2010 (165 days)

        2. Sample C5              Dec. $13^{th}$ – Dec. $22^{nd}$ 2010, Jan. $4^{th}$ – Jan. $8^{th}$, Jan. $12^{th}$ – Jan. $14^{th}$ 2011 (18 days)

        3. Sample S1               Dec. $23^{rd}$ – Dec. $29^{th}$ 2010 (7 days)

4. Sample SC              Dec. $30^{th}$ – Dec $31^{st}$ 2010 (2 days)

        5. Sample S4               Jan. $1^{st}$ – Jan. $3^{rd}$, Jan. $9^{th}$ – Jan. $11^{th}$ 2011 (6 days)

        6. Sample S7               Jan. $15^{th}$ – Jan. $17^{th}$ 2011 (3 days)

        7. Sample 2B              Jan. $18^{th}$ – Jan. $25^{th}$ 2011 (8 days)

        8. Sample BS1            Jan. $26^{th}$ – Feb. $1^{st}$ 2011 (6 days)

9. Sample BS2            Feb. $2^{nd}$ – Feb. $13^{th}$ 2011 (12 days)

        10. Sample BS4         Feb. $14^{th}$ – Mar. $13^{th}$ 2011 (28 days)

        11. Sample BS7         Mar. $14^{th}$ – Mar. $24^{th}$ 2011 (11 days)

        12. Sample BS3         Mar. $25^{th}$ – Apr. $4^{th}$ 2011 (11 days)

        13. Sample C14         Apr. $5^{th}$ – Apr. $17^{th}$ 2011 (13 days)

14. Sample C11B       Apr. $18^{th}$ – Jun. $30^{th}$ 2011 (73 days)

**Table 2**: Monthly net primary productivity of the Boodjamulla biological soil crust (values in brackets are an estimation only, not based on measurements; see text for explanantion).

| Month | NPP (NP-DR) | |
| | (mmol $CO_2$ m$^{-2}$ month$^{-1}$) | (g C m$^{-2}$ month$^{-1}$) |
| --- | --- | --- |
| July | 0 | 0 |
| August | 0 | 0 |
| September 2010 | 0 (-2.0) | 0 (-0.02) |
| October 2010 | 0 (-86.0) | 0 (-1.06) |
| November 2010 | -210.26 | -2.53 |
| December 2010 | 110.20 | 1.32 |
| January 2011 | 99.58 | 1.20 |
| February 2011 | 80.99 | 0.97 |
| March 2011 | 174.11 | 2.09 |
| April 2011 | -25.54 | -0.31 |
| May 2011 | 0 | 0 |
| June 2011 | 0 | 0 |
| **Annual** | **229.09 mmol $CO_2$ m$^{-2}$ yr$^{-1}$** **(143,08 mmol $CO_2$ m$^{-2}$ yr$^{-1}$)** | **2.75 g C m$^{-2}$ yr$^{-1}$** **(1.72 g C m$^{-2}$ yr$^{-1}$)** |

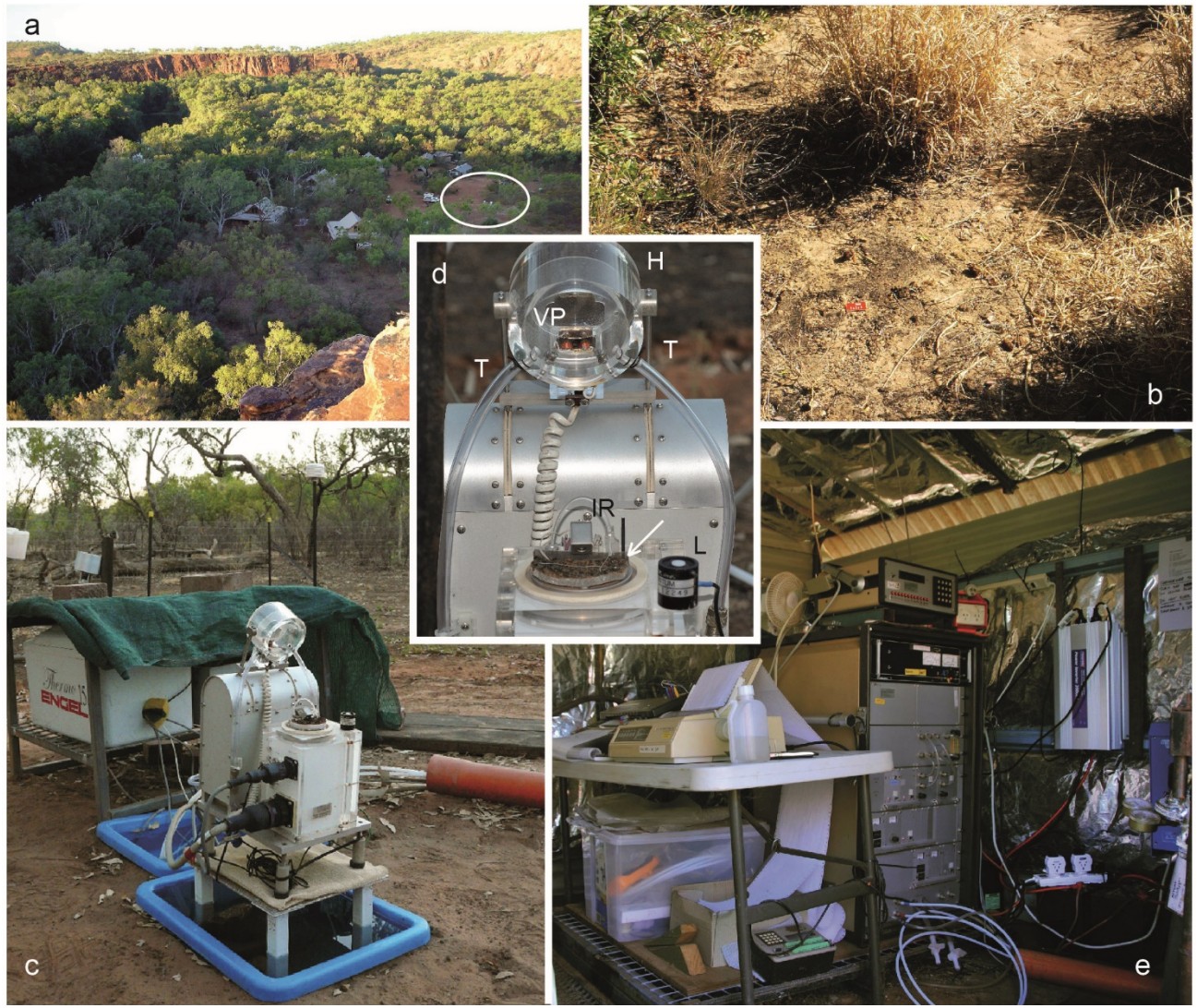

Figure 1

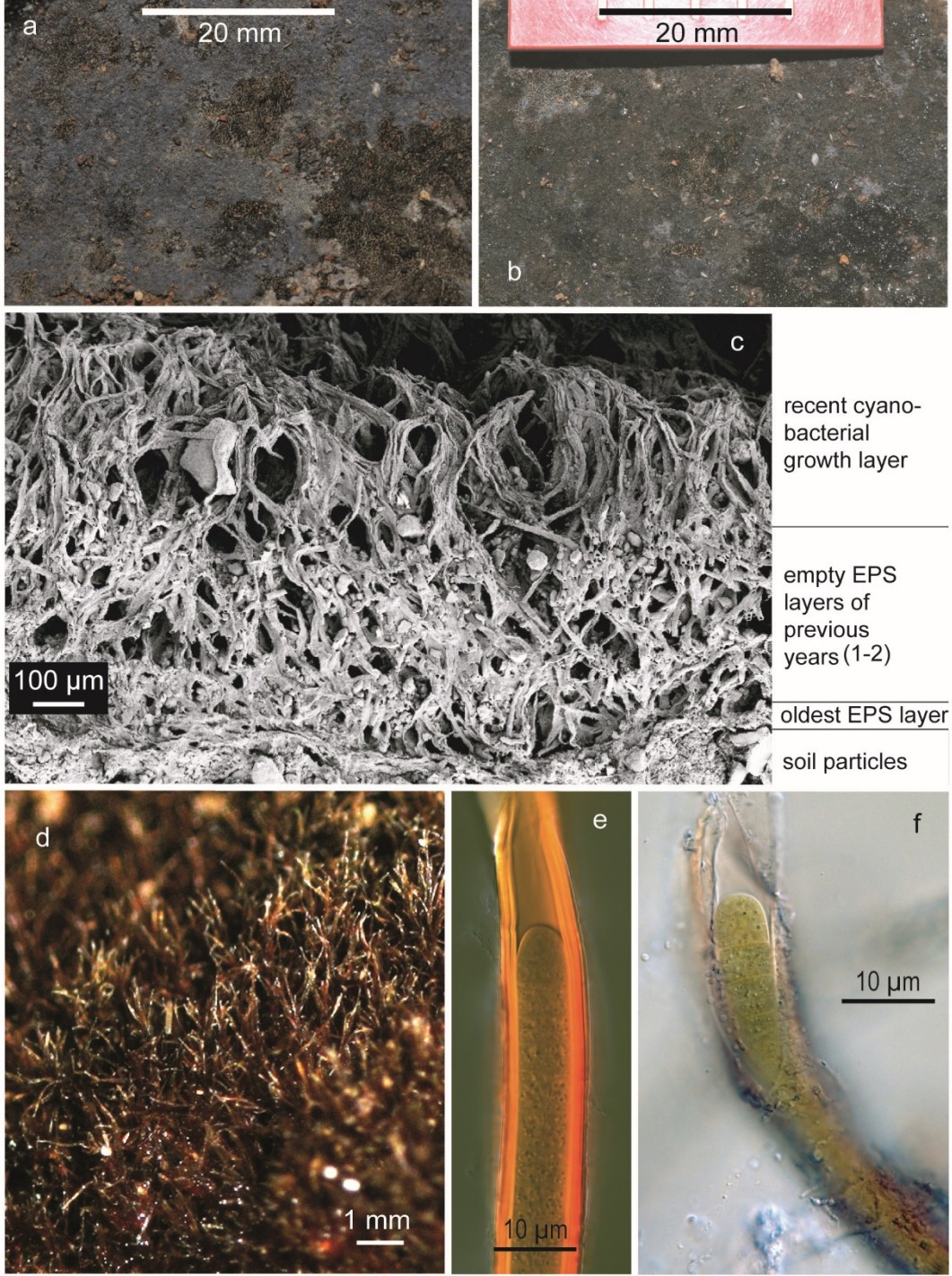

Figure 2

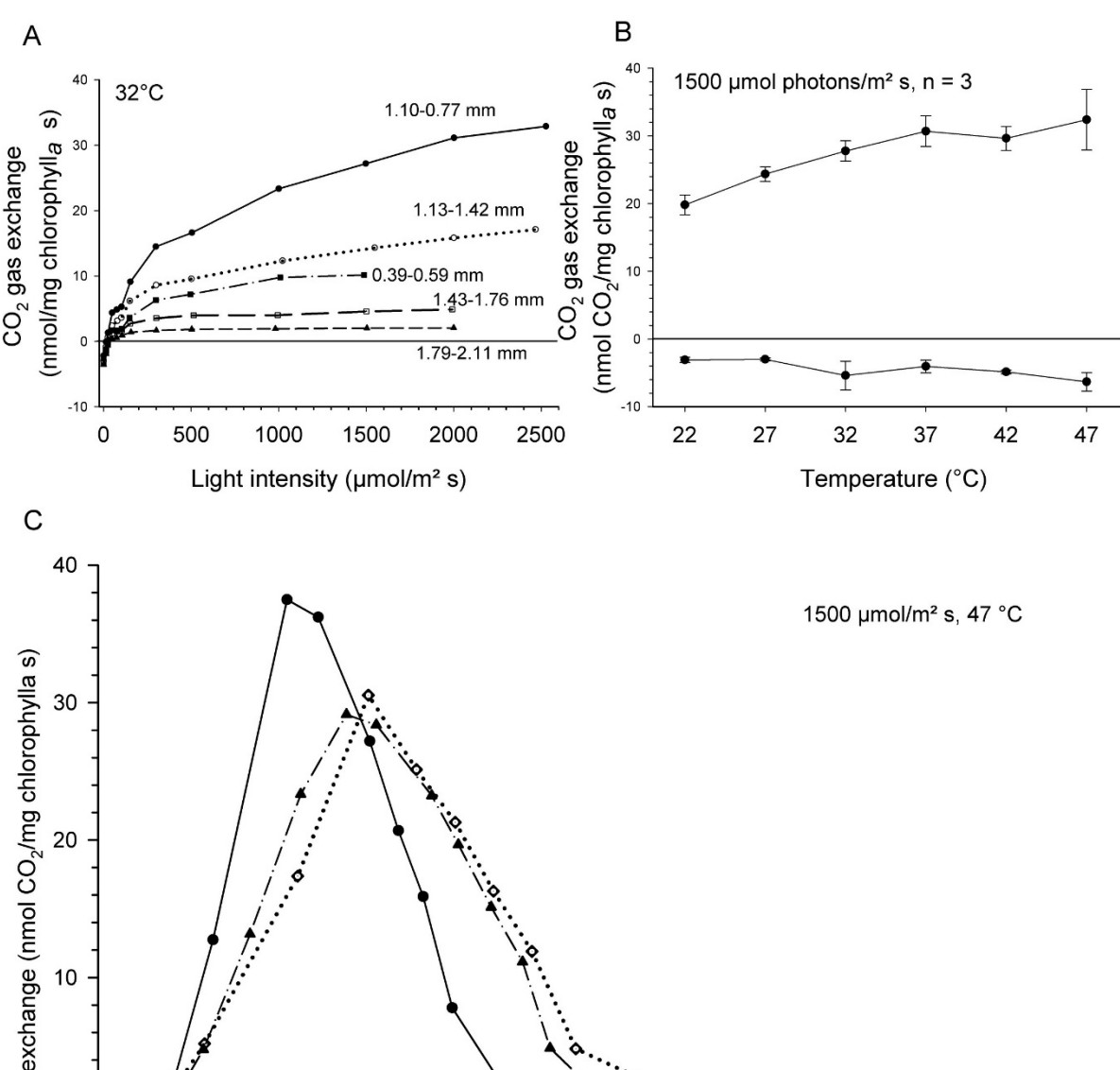

Figure 3

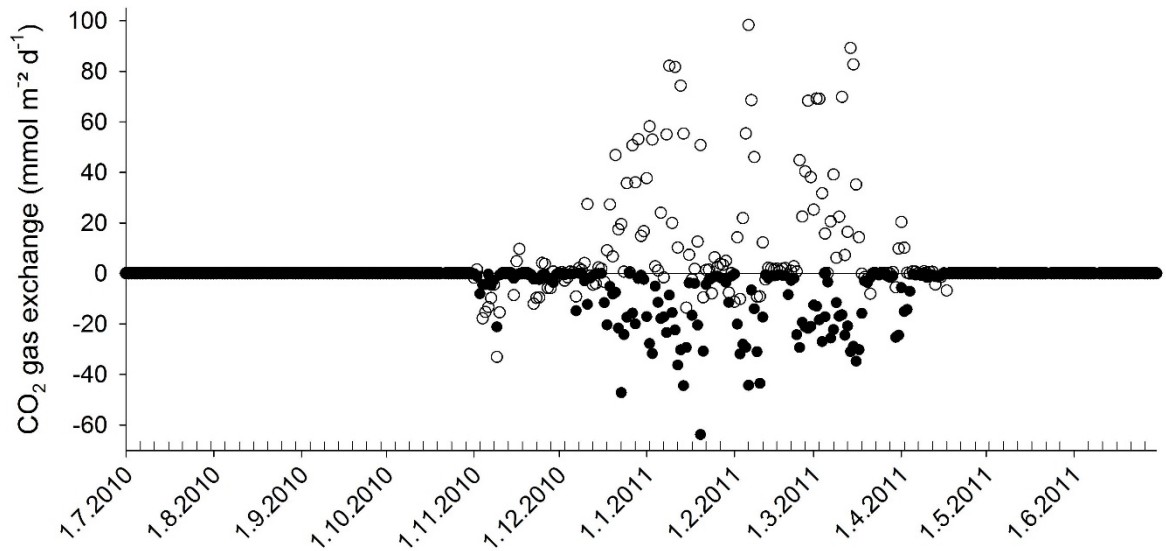

Figure 4

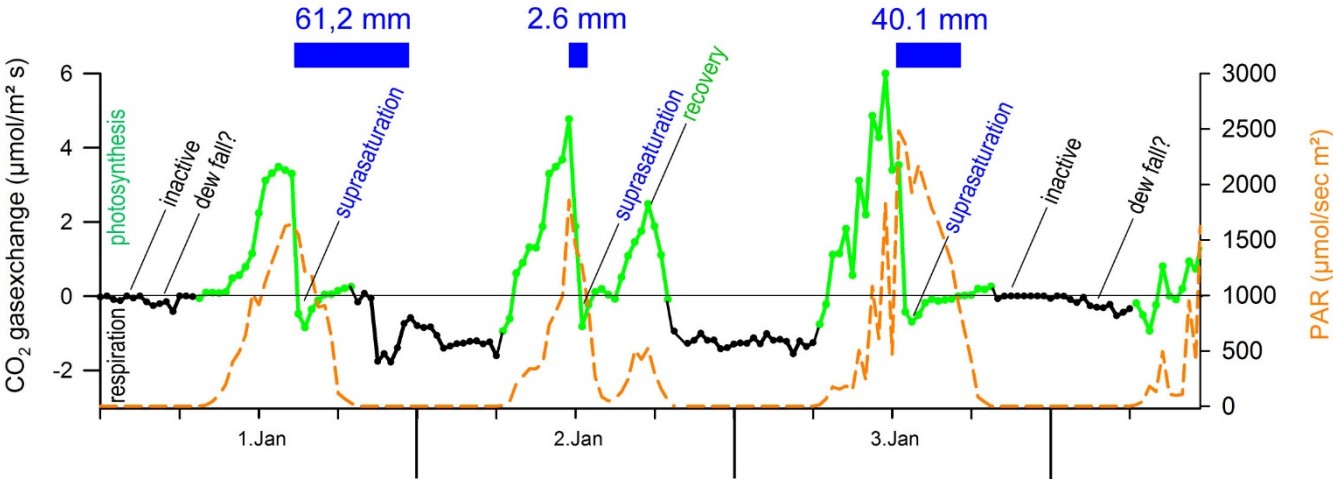

Figure 5

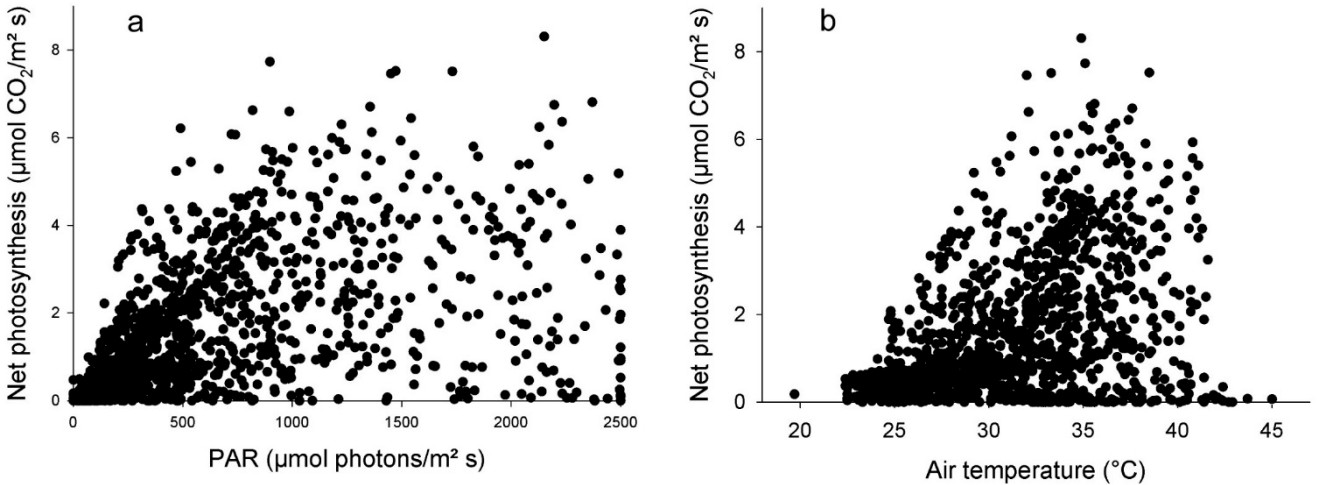

Figure 6

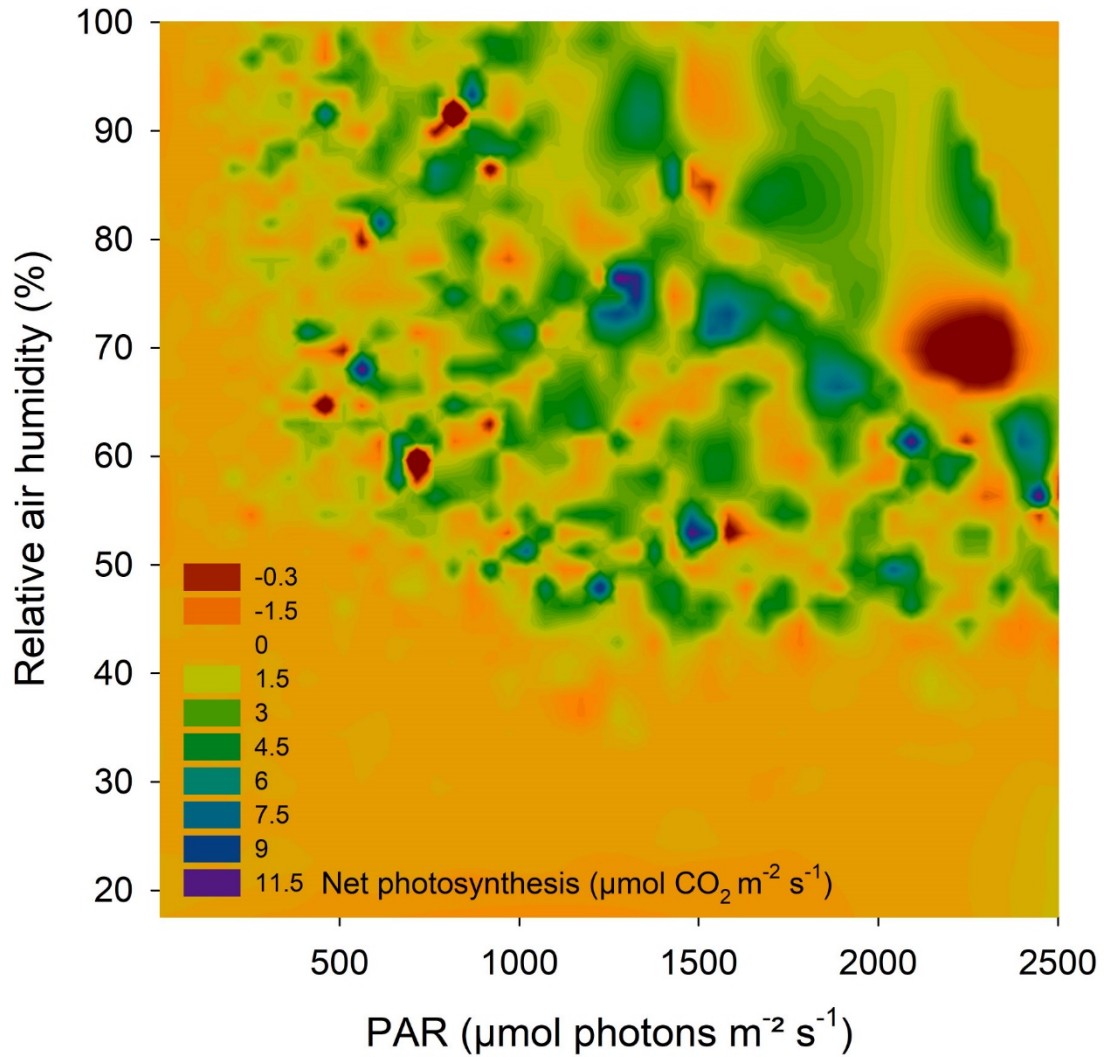

Figure 7

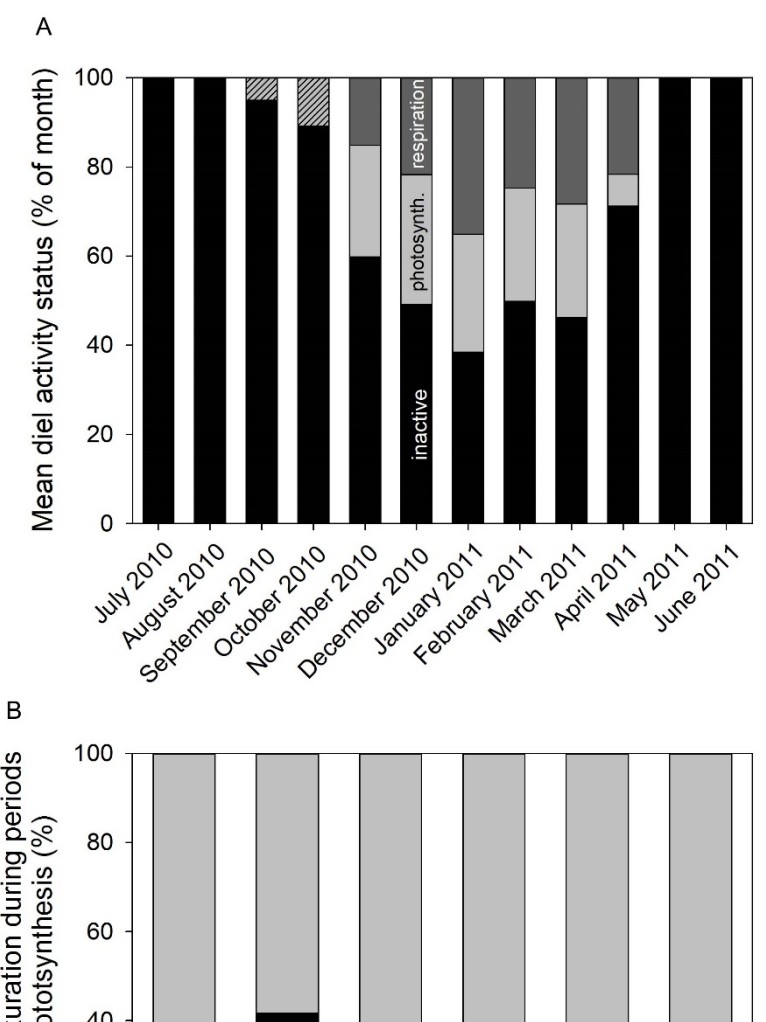

A

Mean diel activity status (% of month)

respiration

photosynth.

inactive

July 2010
August 2010
September 2010
October 2010
November 2010
December 2010
January 2011
February 2011
March 2011
April 2011
May 2011
June 2011

B

Water suprasaturation during periods of net phototsynthesis (%)

suprasaturation

November 2010
December 2010
January 2011
February 2011
March 2011
April 2011

Figure 8

