# Peer review of "Annual net primary productivity of a cyanobacteria-dominated biological soil crust in the Gulf savannah, Queensland, Australia"

_Biogeosciences, 2017_

## Referee Comment (RC1) · M. Bowker (Referee) · 11 Oct 2017

M. Bowker (Referee)

matthew.bowker@nau.edu

Dear Dr. Büdel et al. – I have reviewed your manuscript "Annual net primary productivity of a cyanobacteria dominated biological soil crust in the Gulf savanna, Queensland, Australia", submitted to Biogeosciences. This paper tackles a challenging measurement problem: estimation of the net C flux of a biocrust community over a year in the field. Also, using a battery of controlled environment treatments, the authors determine the response of these biocrusts to moisture, temperature, and light. Overall, the authors find that biocrusts are a net C-sink in this environment, but net production is only observed for a portion of the year. The strength of the paper is that the

authors have amassed an impressive amount of data and are one of only a handful of groups to complete this type of estimate. The weaknesses are perhaps due to a weak expression of why it is so important to conduct this measurement, and why other similar measurements have been scarce, and an occasional propensity to dwell on details without clear explanation of why they are important. Below, I provide several suggestions to help revise the paper.

Major comments: 1. I understand that this study does not fit the typical hypothesis test framework, but nonetheless the authors could ensure that readers comprehend the more interesting elements of this work in the abstract, introduction and throughout. We can be fairly confident that most persistent biocrusts have a positive C-balance, because if they did not they would cease to exist eventually. Readers may find it intriguing that despite this apparent tautology, it is difficult to actually observe net $CO_2$ uptake in biocrusts. This is distressing given that due to their extent, biocrusts may be non-trivial players in the global C cycle today, and almost certainly were major players in early terrestrial communities. We need this information.

The reasons are various, but 2 major ones are that the positive $CO_2$ uptake only occurs during a small part of the year in most studies, and it is difficult to separate C-balance of biocrusts from C-flux from organisms (microbes, roots) or minerals (carbonates) that occur below them. If the study is framed as outlined above, obtaining an annual measurement becomes much more intriguing to the casual reader and the importance of this endeavor is understood.

2. Consider standardizing terminology for the one year monitoring (also called "monitoring of gas exchange") and the factorial experiment (also called "gas exchange under controlled conditions"). I might suggest "environmental manipulations" and "Field monitoring".

3. Consider placing the material on P4 L8-13 in section 2.3, and P4 L14-21 in section 2.4. It might improve flow and understandability.

4. P 4 L9 – Why were the samples stored frozen? This does not seem like a region where freezing soils are natural. Aren't you worried this exposure could have harmed or otherwise altered the samples?

5. I could benefit from a few more details about how the 21 samples were used. For example you say 9 were used for the environmental manipulations, and 11 were used for long-term monitoring. What about the 21st sample? Also, I understand you inserted different biocrust samples for different portions of the field monitoring. But why are the samples used for such wildly varying times, I would have thought each would be used about 1.1 months?

6. There are 10 figures, are they are really needed? The content of Figure 6 is mentioned by the authors several times, but it is not completely clear to me why the authors ascribe so much importance to these 3 days. Also, figures 9 and 10 could probably be combined into one 2 paneled figure.

7. There are times when I would like to see different pieces of information integrated, and another case where there is integration but I do not have all the information I need to understand it. Fig. 4. Provides plots of biocrust responses to different environmental gradients in a manner often used by this author group and associates. This is fine, but what I haven't ever seen is a plot integrating more than one of these variables in 3 dimensions. This would be a nice addition, if it could be done. Fig. 8 is a valiant attempt at illustrating responses to 2 environmental variables as a surface, but there is no explanation of how this was created (Kriging?); further, the plot contains many inexplicable peaks and valleys, often near each other. Does this suggest overfitting? Maybe more aggressive smoothing is warranted.

8. The discussion is not bad as written. You do address a key measurement issue, and hypothesize that the isolation of biocrust samples from underlying soil is the reason some studies find net C-uptake, and some find net C-loss.

I would have like to see you more fully develop a few other elements too (several
of which you do address to some degree), for example the generality that biocrusts maintain their existence by attaining positive C-balance only during a portion of the year, and that often the gains over a year are marginal. This means that oft-cited slow natural growth rates likely are due to environmental constraints; only a minority of the year is actually suitable for growth. I would have like to see you advance some hypotheses for why different regions have different annual C-flux values. Related to this, one novel aspect of your study is that all other annual flux measurements were conducted in environments with cool season hydration. Finally, you could develop more your hypothesis about how expected climate changes might impact these naturally occurring biocrusts. It might be helpful to break the discussion into a few subsections devoted to distinct discussion topics.

Minor comments Throughout: I suggest using "cyanobacterially dominated" (adverb modifying adjective) or "cyanobacteria-dominated" (noun functioning to modify adjective), not "cyanobacteria dominated"(no hyphen, no adverb) P1L18 - remove "at" P1L19 - remove "during", suggest replacement of "referring" with "corresponding" P2L21-23 – standardize terminology for net C-uptake, 3 different synonyms are used here P2L27 – This would be a good place to mention that apparent C-source behavior is probably due to the challenges of properly measuring biocrust C-flux P4L15 – your meaning is unclear in the phrase "making sure that the area related…...range" P4L18 – suggest "drainholes" rather than "borings" P7L1 – suggest "monitoring" rather than "investigation" P7L29 – suggest "continuing" rather than "continued" P8L12 – that biocrusts are typically losing C does not mean that overall they are a C-source. P8L15 – Omit "When"

I hope you find these comments constructive – Matthew Bowker

---

## Referee Comment (RC2) · Anonymous Referee #2 · 11 Oct 2017

GENERAL COMMENTS This is a very nice study providing some data on a topic for which data is quite scarce: a high-temporal-resolution examination of isolated biocrust C exchange on an annual time scale in natural conditions. My main recommendation is to work on the framing of the study and improving the context of the results. These could be more compelling in a couple ways. First, the setting up at the beginning of a "mystery" that some crusts take up C and others don't does not work for the structure of this paper. I already knew the answer before seeing the results that it was a difference in methods among the studies (isolated crusts vs. whole soil column) so the results did not resolve this part of the narrative.

[Figure]

Instead of taking this 'source vs. sink' approach to framing the study, I recommend more directly addressing the contribution of isololated crusts vs. other components of the ecosystem, which though not measured here, are measured much more often than isolated crusts in other studies. This brings me to the context of the results. The order of magnitude of one of the main results here, the 1.72 g m-2 is quite interesting (a very significant finding for which it is hard to come by data). It would be worth putting this in the context of eddy flux tower-measured NEE values, which tend to be on the order of 10s to 100s of g C m-2. For example, a recent study of drylands showed NEP varying from 350 to +330 g C m-2 (Biederman et al. 2017 http://dx.doi.org/10.1111/gcb.13686). Furthermore, the other cited studies that do not isolate crusts report values on the order of 50 g C m-2 losses, and these losses likely come from plant roots (which the authors should discuss).

The results in this study suggest that net exchange from these biocrusts are 1-2 orders of magnitude below the flux magnitudes of plants, possibly suggesting they are not a huge part of the total ecosystem C budgets. The Elbert et al review claims global NPP by cryptogamic covers to be 7% globally but that seems hard to reconcile with this type of result. Regardless of whether the authors totally agree with my reasoning here, this kind of context is worthy of more exploration in lieu of the source vs. sink issue. The true strength of this study, the knowledge gap it fills, is in helping to understand the annual fluxes of an isolated crust into our overall understanding of ecosystem C exchange in systems that contain biocrusts. For that, it is a great study and certainly worthy of publication. The exploration of seasonality and environmental effects on flux rates are also quite nice and well done.

SPECIFIC COMMENTS abstract: "Of the metabolic active period, 48.6% were net photosynthesis and 51.4% dark respiration." 48.6% of what? measurement timepoints? typo: "above a relative humidity above 42%"

Abstract: "This must be taken into consideration for future analyses and modelling of carbon balances in comparable biocrust ecosystems." I think a much more solid

conclusion can be given here. It's not clear what comparable biocrust systems are being referred to and "it might be helpful for modeling" is never very compelling. I would end with direct and clear answers to the questions and what the results mean to the larger study of biocrust C exchange.

Take a look at the significant digits throughout the manuscript. eg 31.66 nmol feels unrealistically and unnecessarily precise. This will make it easier to read too.

P6 L5: missing close parenthesis?

I'm not wild about the reporting of WC as mm precip equivalent. Wouldn't moisture levels depend on both precip in as well as evapotransipiration and drainage out? Why not report as something intuitive such as g water g-1 wet soil (ie % water by mass)? If kept, include an explanation of why this approach was used.

I like figures 1 and 2. I wish more papers would include nice methods photos like this.

Figure 3 caption needs to define what the codes on the horizontal axis (x axis?) mean. I think it should say x axis and it says sample signature, which I didn't understand. Does that mean an individually collected sample? I counted them and see 21 so I am guessing that's what it means. The codes are a bit cryptic.

P2 L 23. what is meant by "carbon deposition"

P2. L25. Part of the issue here is that studies are being mixed in which the biocrusts are by themselves or sitting on top of intact soil in which other fluxes like sub-crust respiration and root respiration contribute. This study uniquely looks at an isolated crust with very high temporal resolution. It is quite interesting that net uptake is on the order of 1g m-2 year. This is quite meaningful when comparing with eddy flux values from drylands and many ecosytems that show NEE values on the order of 10s or 100s of g C m-2 year. This suggests that cyano crusts alone are not huge contributors to C uptake, even in dryland ecosystems. (Note: I jotted this paragraph down before reading the results and discussion)

I'd phrase the crucial question part of the intro more clearly and flag them as such with numbers and questions marks. Something like: This led us to two questions: (1) How do Boodjamulla biocrusts respond to the pronounced seasonality of water availability? and (2) Are they sources or sinks for carbon at the annual timescale?

"So far, it is unclear what is triggering a biocrust as either a sink or a source of C." I'm not sure this is conceptually accurate. If the biocrust itself, isolated as you have done here" is a persistent source, it dies. If it's a sink, it is growing or at least building SOM. I think what's unclear is the role of the crusts in the larger ecosystem and THAT'S what this study shows probably better than any previous study.

P8 L10. A key point here though is that that top soil chamber still includes plants because there are plant roots below those crusts. The same is true for all the chamber approaches, and is one of the key reasons studies of isolated crusts like this one is valuable and needed. Those numbers (48.8 and 62) are likely large because they are catching the flip side of plant photosynthesis via roor respiration, which is a lot higher than what the crusts are doing. Chamber depth is more or less irrelevant because they are not closed on the bottom. Any root or microbe from anywhere below the measurement area would contribute to surface fluxes.

In conclusion, some key take-homes from this study I'd like to see emphasized more are that the cyano crust alone is capable of being a small but consistent sink of carbon as it grows and possibly contributes a bit to SOM. However, it's clear also from the magnitude of the values that we are not seeing C fluxes that are anywhere close to what the plant community can do. A more complete discussion of these quantities would greatly improve the presentation of this excellent data set.

---

## Short Comment (SC1) · 11 Oct 2017

Dear Matthew,

thank your very much for your very helpful comments. I will include and respond to all your excellent comments. Some of them I already had prepared as for example an explanation why cyanobacteria-dominated biocrust's resurrection is largely a regrowth: It seems that lichens provide an environment for cyano- and chlorobionts that allows considerably higher survival rates (near to 100%) for the cyanobacterial-/algal cell than their natural non associated colonies. It is known from Literature that in cyanobacteria colonies only a certain number (5-30%) of cells survive drought periods, depending

on onset, duration and rewetting of droughts. High respiration rates are the result of an increased resurrection. However in lichens the whole consortium is comming back to live and they can start after a few hours or, more rarely, days with positive net photosynthsis.

I will extend the introduction and discussion according to your suggestions and will also try to minimize the number of figures..

Best wishes and thank you once more, Burkhard

---

## Referee Comment (RC3) · Anonymous Referee #3 · 13 Oct 2017

Review report on the manuscript bg-2017-374, authored by Burkhard Büdel et al., titled "Annual net primary productivity of a cyanobacteria dominated biological soil crust in the Gulf savanna, Queensland, Australia", submitted to the journal Biogeosciences, for the Special Issue: Biological soil crusts and their role in biogeochemical processes and cycling

General comments This work deals with the metabolic activity and gas exchange of a cyanobacterial biocrust in the Boodjamulla National Park of Australia. The authors have identified the main species in the biocrust, have carried out lab measurements of net photosynthesis (NP) and dark respiration (DR) as related with variable conditions of

light, temperature and water content (WC), and have carried out one-year monitoring of NP and DR under field conditions, recording also a series of micro-climatic data. The work is very interesting, in the cutting edge of knowledge, fully matches with the scope of the BG special issue, and is well made and well written with almost the only exception of some aspects in the methods section, which has consequences in part of the results. However, I think all my comments can be resolved with a few changes. The results includes the time duration of metabolic activity of the biocrust, the time proportion of NP vs DR, and an explanation of the timing of these processes. It should be noted that the authors provide with annual net CO2 flow data. They also provide evidence of strong seasonality of these biocrust-atmosphere gas exchange processes. and of the positive annual balance of NP despite the relatively short time during which NP is achieved, showing that the net incoming CO2 flux produced by NP is larger than the flux due to DR, since the duration of NP seems shorter. The positive annual balance of CO2 is important because it is not obvious, having into account the low frequency of the conditions in which positive NP can be observed in situ in biocrusts in most of the sites. Discussion and Conclusion are based in the results, include substantial contributions to knowledge and, along with the Introduction, show the experience and up-to-date knowledge of the authors.

Specific comments Methods The sampling is a bit confusing. How many samples/replicates were used for the experiment of CO2 exchange under controlled conditions in lab? It seems that three replicates were used but the sentence in lines 12 and 13 of page 4 introduces some doubts. It is unclear what '9 different samples' are. If I have understood well the main text, I suggest rewriting that sentence, for example: "For every independent variable (light, temperature and water content), a different set of three samples/replicates was used". About the samples used for the one-year monitoring, taking some replicates in each measurement time would have been better. I think that it would been possible with only one cuvette (measurements in a series of replicates can be done in a short enough period to avoid that daily gas exchange variation had a significant effect in part of the replicates with regard to the others). However, Fig 3 suggests (showing the time overlaps among bars of different colours) that sometimes through the year only one sample was measured, whereas in others, two, three, four or even five samples were measured. Were there any replicates at certain times of the year? If this is so, I think that this diversity in number of replicates along the year requires some comment. If this is not the case, the Fig 3 should be corrected or explained. Independently from the number of replicates during the monitoring period, the use of several different samples throughout the year would have been probably necessary, because the cyanobacterial biocrust samples have a limited resistance to handling and, after a series of measurements, they should be replaced. (Due to this fact, this is not properly a case of repeated measures over time). But again Fig 3 shows how the duration of the different samples is very different; in some cases the same sample appears to have been used repeatedly during even five or six weeks, but in others, only once. Is this related with the difficult to understand the last sentence of section 2.2 (page 4, lines 20 - 21)? Please, explain better how, "for the one-year monitoring", you used 21 samples (page 4, line 16) and, though from those 21 used samples, only 11 were selected "for the long-term monitoring" (page 4, lines 20-21). Do you mean that those selected 11 samples were used repeatedly over time while the other 10 were used only once? Why?. By the way, the caption of the Fig 3 should be completed; a caption must be self-explanatory. I think we can assume that replacing the samples along the year for the monitoring is acceptable, apart from possibly necessary, since the monitoring was made only on carefully selected samples of the brown cyanobacterial community dominated by Symplocastrum purpurascens, and each sample includes probably billions of cyanobacterial individuals, being a good representation of the whole community. Besides, the microbiota at a certain sampling point could change enough along the year, which decreases the importance of always sampling at the same point. On the other hand, it would be advisable to state explicitly the number of times in which measurements were taken during the monitoring (and when), avoiding the reader having to speculate or discover this from the figures. For example, writing, "twenty-five measurements were taken between November and April,

СЗ

once per week, on the dates shown in the Fig 3". The sentence (page 4, lines 24-25) "The response of ... NP and ... DR to ... WC was determined for light, temperature and WC" could be better written, to avoid the expression 'the response to WC was determined for... WC'. I am not sure I have understood this paragraph, particularly after seeing Fig 4. According lines 30-31 of page 4, the temperature-related NP and DR were determined by varying temperatures while keeping constant both light and WC; whereas the WC-related NP and DR were determined (lines 1-2 of page 5) at constant light and different temperatures (in addition to different WC, it is supposed). However, according the Fig 4, it seems that temperatures and WC were not crossed. On the other hand, the first step of the procedure was determining the effect on NP and DR of light in every level of WC, for constant optimal temperature. I wonder whether the determination of the effect on NP and DR of temperature by itself (for constant optimum conditions of light and of WC) was the procedure for stablish that optimal temperature. If so, then this should be explicit and constitute the firs step. If not, why is studying the effect of temperature by itself important since the effect on NP and DR of WC was determined for every temperature (keeping constant light)?. I do not think these experiments were badly done, only that this paragraph is difficult to understand and raises doubts. Since to test the effect of each of these independent variables, at least one of them remained constant, the design is not fully factorial. Probably the triple interaction is significant and, in such a case it would be interesting to understand the biocrust functioning under natural conditions, to study the NP and DR response to that triple interaction, rather than the responses to every independent variables more or less separately. Nevertheless, meanwhile, this work provides very valuable information. In page 4, line 31, is 1500  $\mu$ mol photons m2 s-1 a saturating light? And, how is "optimal WC" defined? A (very short) definition appears only much after, in the line 15 of page 6. On the other hand, is the whole procedure described in the last sentence of this paragraph (lines 2-4 of page 5) repeated for every temperature? This is almost obvious but I think that to say it explicitly would be better.

Results The lines in Fig 4 b are not attributed to any WC levels. A series of lines

similar to those of graph from Fig 4a are expected here, if I understood adequately the methods. Caption of Fig 4 does not help to understand this; in the part referred to graph 4b, any reference to the WC levels is missing. If the graph from Fig 4b refers to the effect on NP and DR of the temperature while keeping constant both light and WC, a value of (optimal) WC is lacking in the graph. What are the lines of Fig 4c, has each sample one line of NP and one line of DR? How? Where are the six different temperatures, since in the graph 4b temperatures and WC are not crossed?. Why the graph 4c shows 47°C as constant? experimental temperature whereas, according the main text (page 5, lines 1-2), six temperature were crossed with different WC levels?. Why was the graphed experiment made at  $47^{\circ}$ C instead of at the optimal temperature (32°C)?. Perhaps the authors plotted a graph for every temperature and only show the last one; but, in such a case, what are the lines of Fig 4c?. The wording of Methods and/or Results should be a bit improved.

Conclusion In page 10, line 29, the sentence "three months having a negative balance probably due to regrowth of the biocrust" is hard to understand since, by default, 'regrowth' implies growth, and growth requires net CO2 assimilation. Besides, I think that this sentence about the regrowth requires an explanation, defining what exactly means 'regrowth' in this case, since this is closely related with the hypothesis presented at the end of the introduction (page 3, lines 12-14). Perhaps the authors used here the word 'regrowth' to refer to the recovery of metabolic activity after the latent-life span of the dry season. It would be also advisable a better definition of that hypothesis in the Introduction

Technical corrections Page 4, line 2; the expression '(factorial design)' would be better than '(factorial analysis)'. Indeed, the experiment was factorial (although not fully factorial); but, no statistical analysis is explicit in the Method section. Page 4, line 19: 'NP' and 'DR' appear in that line, whereas they are defined after, in lines 24 and 25 of that page 4 Page 5, line 27: "2" is lacking after 'Fig' Page 6, line 15: A dot or a semicolon seems advisable just before the last word of that line.

---

## Editor Comment (EC1) · E. Rodriguez-Caballero (Editor) · 16 Oct 2017

Dear Büdel et al.,

All three referees acknowledge the potential interest of your work, but between them, they also raise a number of concerns. Please carefully revise your manuscript accordingly taken into account these comments and incorporates the necessary changes. Please provide a version in which the modifications are highlighted, and another one with the modifications accepted. Sincerely

Emilio

---

## Short Comment (SC2) · 6 Nov 2017

Response to Referee 1 (M. Bowker) This paper tackles a challenging measurement problem: estimation of the net C flux of a biocrust community over a year in the field. Also, using a battery of controlled environment treatments, the authors determine the response of these biocrusts to moisture, temperature, and light. Overall, the authors find that biocrusts are a net C-sink in this environment, but net production is only observed for a portion of the year. The strength of the paper is that the authors have amassed an impressive amount of data and are one of only a handful of groups to complete this type of estimate. The weaknesses are perhaps due to a weak expres-

sion of why it is so important to conduct this measurement, and why other similar measurements have been scarce, and an occasional propensity to dwell on details without clear explanation of why they are important. Below, I provide several suggestions to help revise the paper.

Answer: Thanks for the positive valuation. We tried hard to eliminate the weaknesses in expression and also to explain every detail and why they are important.

Major comments: 1. I understand that this study does not fit the typical hypothesis test framework, but nonetheless the authors could ensure that readers comprehend the more interesting elements of this work in the abstract, introduction and throughout. We can be fairly confident that most persistent biocrusts have a positive C-balance, because if they did not they would cease to exist eventually. Readers may find it intriguing that despite this apparent tautology, it is difficult to actually observe net $CO_2$ uptake in biocrusts. This is distressing given that due to their extent, biocrusts may be non-trivial players in the global C cycle today, and almost certainly were major players in early terrestrial communities. We need this information. The reasons are various, but 2 major ones are that the positive $CO_2$ uptake only occurs during a small part of the year in most studies, and it is difficult to separate C-balance of biocrusts from C-flux from organisms (microbes, roots) or minerals (carbonates) that occur below them. If the study is framed as outlined above, obtaining an annual measurement becomes much more intriguing to the casual reader and the importance of this endeavor is understood.

Answer: We rephrased the referring parts of the abstract, introduction and discussion and tried to make the aims and outcomes unambiguously clear throughout the text. We included every suggestion of the referee and hope we could clarify the unclear or weakly expressed parts.

2. Consider standardizing terminology for the one year monitoring (also called "monitoring of gas exchange") and the factorial experiment (also called "gas exchange under controlled conditions"). I might suggest "environmental manipulations" and "Field monitoring".

Accepted, we changed this in the manuscript.

3. Consider placing the material on P4 L8-13 in section 2.3, and P4 L14-21 in section 2.4. It might improve flow and understandability.

Accepted.

4. P 4 L9 – Why were the samples stored frozen? This does not seem like a region where freezing soils are natural. Aren't you worried this exposure could have harmed or otherwise altered the samples?

Answer: we added the following information for better understanding. "….This treatment had been tested in our laboratory many times with lichens of many different geographical origins, including the tropics and resulted in high survival rates (roughly 95%) compared to dry storing in herbarium cabinets or boxes in the laboratory. Earlier gas exchange measurements on biocrusts, cyanobacteria, byrophytes and lichens before freezing and after thawing and re-moistening resulted in identical rates (unpublished laboratory tests)".

5. I could benefit from a few more details about how the 21 samples were used. For example you say 9 were used for the environmental manipulations, and 11 were used for long-term monitoring. What about the 21st sample? Also, I understand you inserted different biocrust samples for different portions of the field monitoring. But why are the samples used for such wildly varying times, I would have thought each would be used about 1.1 months?

Answer: We understood that this graph was confusing, as well as the text. We replaced the figure by Table 1, where we give just the sample and from when to when it was used. We also explained the somehow chaotic seeming randomization of the sample changing mode using the following sentence: "Fourteen samples were used during field monitoring (Table 1) and exposed in a random mode. The "random" mode was

determined by the ability of access (climatic conditions, days off) by one of us to the investigation site during the whole measuring period".

6. There are 10 figures, are they are really needed? The content of Figure 6 is mentioned by the authors several times, but it is not completely clear to me why the authors ascribe so much importance to these 3 days. Also, figures 9 and 10 could probably be combined into one 2 panelled figure.

Answer: We omitted figure 3 and replaced it by a much clearer table. Figures 9 and 10 were combined as suggested so that the number of figures is now reduced to 8.

7. There are times when I would like to see different pieces of information integrated, and another case where there is integration but I do not have all the information I need to understand it. Fig. 4. Provides plots of biocrust responses to different environmental gradients in a manner often used by this author group and associates. This is fine, but what I haven't ever seen is a plot integrating more than one of these variables in 3 dimensions. This would be a nice addition, if it could be done. Fig. 8 is a valiant attempt at illustrating responses to 2 environmental variables as a surface, but there is no explanation of how this was created (Kriging?); further, the plot contains many inexplicable peaks and valleys, often near each other. Does this suggest overfitting? Maybe more aggressive smoothing is warranted.

Answer: Figure 8, which is now figure 7 is a contour plot made with the SigmaPlot Software and is based on a linear interpolation between measuring points (the same as in line graphs). We changed the figure legend in order to make it more understandable for readers. Each and every data point of net photosynthesis (measurements at daylight) was related with the referring air humidity and amount of light at the time of measurement. The colour indicates $CO_2$ uptake rates (positive) or $CO_2$ loss rates during the day (negative values) or inactivity (0; yellow colour). Here the new legend of figure 8 (formerly 7): "Contour plot of net photosynthesis of the Boodjamulla biocrust based on linear interpolation between measured values. Shown is the active period

from November 2010 to April 2011. Net photosynthesis is related to relative air humidity and photosynthetic active radiation (PAR). No dark respiration values shown! Colour key: yellow = no activity, orange to red = CO2 loss during the day (suprasaturation), light green to violet = CO2 uptake". We find this type of presentation impressive as it really shows how many wet up and dry down cycles a biocrust experiences in its natural environment and how common suprasaturation is.

8. The discussion is not bad as written. You do address a key measurement issue, and hypothesize that the isolation of biocrust samples from underlying soil is the reason some studies find net C-uptake, and some find net C-loss. I would have like to see you more fully develop a few other elements too (several of which you do address to some degree), for example the generality that biocrusts maintain their existence by attaining positive C-balance only during a portion of the year, and that often the gains over a year are marginal. This means that oft-cited slow natural growth rates likely are due to environmental constraints; only a minority of the year is actually suitable for growth. I would have like to see you advance some hypotheses for why different regions have different annual C-flux values. Related to this, one novel aspect of your study is that all other annual flux measurements were conducted in environments with cool season hydration. Finally, you could develop more your hypothesis about how expected climate changes might impact these naturally occurring biocrusts. It might be helpful to break the discussion into a few subsections devoted to distinct discussion topics.

Answer: The discussion is more or less newly written and also separated into several subsections. Here the new discussion: 4. Discussion 4.1 Seasonality and CO2 
[revised manuscript text omitted]

Minor comments Throughout: I suggest using "cyanobacterially dominated" (adverb modifying adjective) or "cyanobacteria-dominated" (noun functioning to modify adjective), not "cyanobacteria dominated"(no hyphen, no adverb)

Accepted, all changed to "Cyanobacterially dominated"

P1L18 - remove "at"

Done

P1L19 - remove "during", suggest replacement of "referring" with "corresponding"

Done

P2L21-23 – standardize terminology for net C-uptake, 3 different synonyms are used here

Done

P2L27 – This would be a good place to mention that apparent C-source behavior is probably due to the challenges of properly measuring biocrust C-flux

Accepted and included

P4L15 – your meaning is unclear in the phrase "making sure that the area related..range"

Taken into regard and replaced by a new sentence: "All samples used were tested for a comparative large NP and DR rate under the given environmental conditions for two measurements (1 hour) in the cuvette system and only those were used that had more or less identical NP and DR rates".

P4L18 – suggest "drainholes" rather than "borings"

Done

P7L1 – suggest "monitoring" rather than "investigation" Done

P7L29 – suggest "continuing" rather than "continued"

Done

P8L12 – that biocrusts are typically losing C does not mean that overall they are a C-source.

Accepted and expressed accordingly

P8L15 – Omit "When"

Done

Thank you Matthew for your very helpful comments.

Relative air humidity (%)

PAR (µmol photons m$^{-2}$ s$^{-1}$)

-0.3
-1.5
0
1.5
3
4.5
6
7.5
9
11.5  Net photosynthesis (µmol CO$_2$ m$^{-2}$ s$^{-1}$)

**Fig. 1.** New figure 7

---

## Short Comment (SC3) · 6 Nov 2017

Anonymous referee 2 GENERAL COMMENT: This is a very nice study providing some data on a topic for which data is quite scarce: a high-temporal-resolution examination of isolated biocrust C exchange on an annual time scale in natural conditions. My main recommendation is to work on the framing of the study and improving the context of the results. These could be more compelling in a couple ways. First, the setting up at the beginning of a "mystery" that some crusts take up C and others don't does not work for the structure of this paper. I already knew the answer before seeing the results that it was a difference in methods among the studies (isolated crusts vs. whole soil

column) so the results did not resolve this part of the narrative. Instead of taking this 'source vs. sink' approach to framing the study, I recommend more directly addressing the contribution of isololated crusts vs. other components of the ecosystem, which though not measured here, are measured much more often than isolated crusts in other studies. This brings me to the context of the results. The order of magnitude of one of the main results here, the 1.72 g m-2 is quite interesting (a very significant finding for which it is hard to come by data). It would be worth putting this in the context of eddy flux tower-measured NEE values, which tend to be on the order of 10s to 100s of g C m-2. For example, a recent study of drylands showed NEP varying from 350 to +330 g C m-2 (Biederman et al. 2017 http://dx.doi.org/10.1111/gcb.13686).

Answer: Yes, you are right and we agreed to take a somehow other framing. With the comments of the two other referees, we now used two clearly addressed aims (as also suggested by you): 1) how do the cyanobacterially dominated biocrusts of Boodjamulla respond to the pronounced seasonality of water availability? and 2) are they sources or sinks for carbon at an annual timescale? We also discussed our results in relation to the outcome of the Biederman et al study: "In a recent study using the Eddy co-variance method, Biederman et al. (2017) found a wide range of carbon sink/source function with mean annual net ecosystem productivity (NEP) varying from -350 to +330 g C m-2 across sites with diverse vegetation types in the dryland ecosystems of south-western North America using evapotranspiration (ET) as a proxy for annual ecosystem water availability. Gross ecosystem productivity (GEP) and ecosystem respiration (Reco) were negatively related to temperature, both interannually within sites and spatially across sites and sites demonstrated a coherent response of GEP and NEP to anomalies in annual ET. Their investigation sites included one region having a noteworthy biocrust cover not accompanied by a dense vascular plant vegetation, the La Paz region of Baja California with an annual C-uptake (NEP) of roughly 90 g m-2. Approximating annual C gain based on the maximal CO2 uptake rates of four biocrust types composed of cyanobacteria, cyanolichens and chlorolichens measured by Büdel et al. (2013) from Baja California, we approach an annual C gain of those biocrusts

of 11 $\pm$ 4 g m-2 (calculation based on 90 active days per year with 34 of them having a sub-optimal $CO_2$ uptake rate of only 25% of maximum due to suprasaturation. Daily rates were calculated by maximum NP for 5 hours per day minus 10 hours R + DR). This is 6.5 times more than our pure cyanobacterially dominated biocrust from Boodjamulla but still 8 time less than found for the Baja California site in the study of Biederman at al. (2017). It could well be that later successional biocrusts with a wealth of different species groups, including bryophytes, lichens and green algae besides of cyanobacteria, might reach even higher annual carbon fixation rates".

Furthermore, the other cited studies that do not isolate crusts report values on the order of 50 g C m-2 losses, and these losses likely come from plant roots (which the authors should discuss).

Done

The results in this study suggest that net exchange from these biocrusts are 1-2 orders of magnitude below the flux magnitudes of plants, possibly suggesting they are not a huge part of the total ecosystem C budgets. The Elbert et al review claims global NPP by cryptogamic covers to be 7% globally but that seems hard to reconcile with this type of result. Regardless of whether the authors totally agree with my reasoning here, this kind of context is worthy of more exploration in lieu of the source vs. sink issue.

Answer: We added the following paragraph: "Considering the Boodjamulla biocrust as a mid-successional type, where an increase in carbon gain might be expected in the future when lichens and bryophytes establish to form a later successional soil crust. Dealing here with a cyanobacterially dominated biocrust of a mid-successional type might explain the low C-uptake rates and also the seeming discrepancy to the values calculated for the global NPP by cryptogamic covers of Elbert et al. (2012)".

The true strength of this study, the knowledge gap it fills, is in helping to understand the annual fluxes of an isolated crust into our overall understanding of ecosystem C exchange in systems that contain biocrusts. For that, it is a great study and certainly

worthy of publication. The exploration of seasonality and environmental effects on flux rates are also quite nice and well done.

Thank you very much for this nice comment

SPECIFIC COMMENTS abstract: "Of the metabolic active period, 48.6% were net photosynthesis and 51.4% dark respiration." 48.6% of what? measurement timepoints?

Answer: we rephrased that sentence for more clarity

typo: "above a relative humidity above 42%"

Done

Abstract: "This must be taken into consideration for future analyses and modelling of carbon balances in comparable biocrust ecosystems." I think a much more solid conclusion can be given here. It's not clear what comparable biocrust systems are being referred to and "it might be helpful for modeling" is never very compelling. I would end with direct and clear answers to the questions and what the results mean to the larger study of biocrust C exchange.

Answer: You are right, we omitted this sentence and tried to give a more meaningful outlook in the conclusions: "From the magnitude of values it is clear that the observed C fluxes are not at all close to what a plant community can do. Methodological approaches analysing the carbon cycling of biocrusts need to critically reflect, that including or excluding sub-biocrust partitions might influence the status of the biocrust as being either considered as a sink or a source. There is an urgent need for more long-term measurements on different biological soil crust types and developmental stages from all climatic regions of the world".

Take a look at the significant digits throughout the manuscript. eg 31.66 nmol feels unrealistically and unnecessarily precise. This will make it easier to read too.

Done

P6 L5: missing close parenthesis?

Done

I'm not wild about the reporting of WC as mm precip equivalent. Wouldn't moisture levels depend on both precip in as well as evapotransipiration and drainage out? Why not report as something intuitive such as g water g-1 wet soil (ie % water by mass)? If kept, include an explanation of why this approach was used.

Answer: We added the following sentences in Material and Methods to explain why we use mm water column instead of g water per g biocrust: "Water content of samples of the experimental manipulations and those of the field monitoring is always expressed as millimeter water column. As it was impossible to remove the sample after each measurement from the monitoring cuvette system to determine the fresh weight corresponding with the measured value by weighing it with a balance (measurements every half an hour, nobody of us could stand at the site for the whole period), the only method getting matchable values between field monitoring and controlled experiments is to express it like rainfall in millimeters water column".

I like figures 1 and 2. I wish more papers would include nice methods photos like this.

Thanks for the nice comment

Figure 3 caption needs to define what the codes on the horizontal axis (x axis?) mean. I think it should say x axis and it says sample signature, which I didn't understand. Does that mean an individually collected sample? I counted them and see 21 so I am guessing that's what it means. The codes are a bit cryptic.

Answer: we omitted figure 3 as it was by far too unclear. All three referees had problems with it and we no replaced by the much easier to understand table 1.

P2 L 23. what is meant by "carbon deposition"

Answer: should be C-uptake, we changed that in the sentence.

P2. L25. Part of the issue here is that studies are being mixed in which the biocrusts are by themselves or sitting on top of intact soil in which other fluxes like sub-crust respiration and root respiration contribute. This study uniquely looks at an isolated crust with very high temporal resolution. It is quite interesting that net uptake is on the order of 1g m-2 year. This is quite meaningful when comparing with eddy flux values from drylands and many ecosytems that show NEE values on the order of 10s or 100s of g C m-2 year. This suggests that cyano crusts alone are not huge contributors to C uptake, even in dryland ecosystems. (Note: I jotted this paragraph down before reading the results and discussion) I'd phrase the crucial question part of the intro more clearly and flag them as such with numbers and questions marks. Something like: This led us to two questions: (1) How do Boodjamulla biocrusts respond to the pronounced seasonality of water availability? and (2) Are they sources or sinks for carbon at the annual timescale?

Answer: thank you for this really good suggestion that we accepted and included.

"So far, it is unclear what is triggering a biocrust as either a sink or a source of C." I'm not sure this is conceptually accurate. If the biocrust itself, isolated as you have done here" is a persistent source, it dies. If it's a sink, it is growing or at least building SOM. I think what's unclear is the role of the crusts in the larger ecosystem and THAT'S what this study shows probably better than any previous study.

Answer: we agree and changed this part too.

P8 L10. A key point here though is that that top soil chamber still includes plants because there are plant roots below those crusts. The same is true for all the chamber approaches, and is one of the key reasons studies of isolated crusts like this one is valuable and needed. Those numbers (48.8 and 62) are likely large because they are catching the flip side of plant photosynthesis via roor respiration, which is a lot higher than what the crusts are doing. Chamber depth is more or less irrelevant because they are not closed on the bottom. Any root or microbe from anywhere below the

measurement area would contribute to surface fluxes.

Answer: we agree and tried to express this in the Discussion.

In conclusion, some key take-homes from this study I'd like to see emphasized more are that the cyano crust alone is capable of being a small but consistent sink of carbon as it grows and possibly contributes a bit to SOM. However, it's clear also from the magnitude of the values that we are not seeing C fluxes that are anywhere close to what the plant community can do. A more complete discussion of these quantities would greatly improve the presentation of this excellent data set.

Answer: Again, we fully agree and tried to express that accordingly.

Thank you for your helpfull comments

[Figure]

**Table 1**: Samples used for monitoring (only those listed used during the active period).

| | |
|---|---|
| 1. Sample A10 | Sep. 23$^{st}$ – Dec. 12$^{th}$ 2010 (80 days) |
| 2. Sample C5 | Dec. 13$^{th}$ – Dec. 22$^{nd}$ 2010, Jan. 4$^{th}$ – Jan. 8$^{th}$, Jan. 12$^{th}$ – Jan. 14$^{th}$ 2011 (18 days) |
| 3. Sample S1 | Dec. 23$^{rd}$ – Dec. 29$^{th}$ 2010 (7 days) |
| 4. Sample SC | Dec. 30$^{th}$ – Dec 31$^{st}$ 2010 (2 days) |
| 5. Sample S4 | Jan. 1$^{st}$ – Jan. 3$^{rd}$, Jan. 9$^{th}$ – Jan. 11$^{th}$ 2011 (6 days) |
| 6. Sample S7 | Jan. 15$^{th}$ – Jan. 17$^{th}$ 2011 (3 days) |
| 7. Sample 2B | Jan. 18$^{th}$ – Jan. 25$^{th}$ 2011 (8 days) |
| 8. Sample BS1 | Jan. 26$^{th}$ – Feb. 1$^{st}$ 2011 (6 days) |
| 9. Sample BS2 | Feb. 2$^{nd}$ – Feb. 13$^{th}$ 2011 (12 days) |
| 10. Sample BS4 | Feb. 14$^{th}$ – Mar. 13$^{th}$ 2011 (28 days) |
| 11. Sample BS7 | Mar. 14$^{th}$ – Mar. 24$^{th}$ 2011 (11 days) |
| 12. Sample BS3 | Mar. 25$^{th}$ – Apr. 4$^{th}$ 2011 (11 days) |
| 13. Sample C14 | Apr. 5$^{th}$ – Apr. 17$^{th}$ 2011 (13 days) |
| 14. Sample C11B | Apr. 18$^{th}$ – Apr. 20$^{th}$ 2011 (3 days) |

**Fig. 1.** Table 1

---

## Short Comment (SC4) · 6 Nov 2017

Anonymous referee 3 GENERAL COMMENTS This work deals with the metabolic activity and gas exchange of a cyanobacterial biocrust in the Boodjamulla National Park of Australia. The authors have identified the main species in the biocrust, have carried out lab measurements of net photosynthesis (NP) and dark respiration (DR) as related with variable conditions of light, temperature and water content (WC), and have carried out one-year monitoring of NP and DR under field conditions, recording also a series of micro-climatic data. The work is very interesting, in the cutting edge of knowledge, fully matches with the scope of the BG special issue, and is well made and well writ-

ten with almost the only exception of some aspects in the methods section, which has consequences in part of the results. However, I think all my comments can be resolved with a few changes. The results includes the time duration of metabolic activity of the biocrust, the time proportion of NP vs DR, and an explanation of the timing of these processes. It should be noted that the authors provide with annual net $CO_2$ flow data. They also provide evidence of strong seasonality of these biocrust-atmosphere gas exchange processes, and of the positive annual balance of NP despite the relatively short time during which NP is achieved, showing that the net incoming $CO_2$ flux produced by NP is larger than the flux due to DR, since the duration of NP seems shorter. The positive annual balance of $CO_2$ is important because it is not obvious, having into account the low frequency of the conditions in which positive NP can be observed in situ in biocrusts in most of the sites. Discussion and Conclusion are based in the results, include substantial contributions to knowledge and, along with the Introduction, show the experience and up-to-date knowledge of the authors.

Thank you for the positive valuation.

SPECIFIC COMMENTS Methods The sampling is a bit confusing. How many samples/replicates were used for the experiment of $CO_2$ exchange under controlled conditions in lab? It seems that three replicates were used but the sentence in lines 12 and 13 of page 4 introduces some doubts. It is unclear what '9 different samples' are. If I have understood well the main text, I suggest rewriting that sentence, for example: "For every independent variable (light, temperature and water content), a different set of three samples/replicates was used". About the samples used for the one-year monitoring, taking some replicates in each measurement time would have been better. I think that it would been possible with only one cuvette (measurements in a series of replicates can be done in a short enough period to avoid that daily gas exchange variation had a significant effect in part of the replicates with regard to the others).

Answer: you are right, that was confusing and we rephrased the whole part as follows: "2.3 Environmental manipulations For the analysis of the effect of the different

environmental factors (light-, temperature- and water content; termed environmental manipulations throughout the text) on net photosynthesis, samples were air dried in a 10 cm Petri-dish, sealed and transported to the laboratory, where they were stored frozen (-20°C) until used for the measurements. This treatment had been tested in our laboratory many times with lichens of many different geographical origins, including the tropics and resulted in high survival rates (roughly 95%) compared to dry storing in herbarium cabinets or boxes in the laboratory. Earlier gas exchange measurements on biocrusts, cyanobacteria, byrophytes and lichens before freezing and after thawing and re-moistening resulted in identical rates (unpublished laboratory tests). Prior to the measurements, samples were allowed to defrost at 23°C for 12 h in an air tight box at low light intensities (« 50 $\mu$mol photons m-2 s-1) in order to avoid decondensation. Subsequently samples passively dehydrated and were kept at 23 °C and natural day-night cycles ($\sim$150 $\mu$mol photons m-2 s-1) for 2 days. Light-, temperature- and water content related NP-measurements were performed using three independent replicates each. $CO_2$ gas exchange measurements were conducted under controlled laboratory conditions using minicuvette systems (CMS 400 and GFS 3000, Walz Company, Effeltrich, Germany). The response of net photosynthesis (NP) and dark respiration (DR) was determined independently for light, temperature and water content (WC). Samples were weighed between measurements and WC was calculated later on as mm precipitation equivalent after final determination of the samples dry weight (exposed 5 days in a desiccator over silica gel at the end of the measurement). To obtain the NP response to light, fully hydrated samples (n=3) were exposed to stepwise increasing photosynthetic active radiation (PAR) from 0 to 2500 $\mu$mol photons m-2 s-1, near optimal temperature (32°C) and ambient $CO_2$ concentration. The light cycle (about 30 min duration) was repeated until the samples were completely dry (after 3–4 h). Light saturation was defined as the PAR at 90% of maximum NP. The temperature related NP and DR were determined at increasing temperature steps, 22, 27, 32, 37, 42, and 47°C, while light was constantly at 1500 $\mu$mol photons m-2 s-1 and WC was constantly at optimum (n=3). The influence of WC on NP and DR was determined at constant,

nearly saturating light (1500 $\mu$mol photons m-2 s-1) and six different temperatures (22, 27, 32, 37, 42 and 47°C) using three replicates. Samples were completely soaked with water and exposed in the cuvette. Then, NP and DR were measured in short time intervals (roughly 10 minutes) until the samples were almost dry and did not show any NP nor DR reactions. After each time interval, the fresh weight of the sample was determined using a balance and the corresponding WC to each data point calculated using the dry weight of the sample (see above). In all experimental manipulations, the $CO_2$ exchange rates of the sample were related to chlorophyll a content. For chlorophyll determination, the samples were ground to small pieces and then extracted two times with di-methyl-sulfoxide (DMSO) at 60 °C for 90 minutes. The chlorophyll a + b content was determined and calculated according to (Ronen and Galun, 1984). 2.4 Field monitoring of $CO_2$ gas exchange As there was only one semi-automatic cuvette system available, we could not replicate the measurements. To partly overcome this problem, we used several samples over the year. Samples were placed in a basket of thermoplastic resin with drainholes in the bottom to avoid standing water during rain events. The basket had a fixed size and all samples had exactly the same exposed surface of 16.5 cm$^2$ (Fig. 1 d). All samples used were tested for a comparative large NP and DR rate under the given environmental conditions for two measurements (1 hour) in the cuvette system and only those were used that had more or less identical NP and DR rates. Fourteen samples were used during field monitoring (Table 1) and exposed in a random mode. The "random" mode was determined by the ability of access (climatic conditions, days off) by one of us to the investigation site during the whole measuring period. Water content of samples of the experimental manipulations and those of the field monitoring is always expressed as millimeter water column. As it was impossible to remove the sample after each measurement from the monitoring cuvette system to determine the fresh weight corresponding with the measured value by weighing it with a balance (measurements every half an hour, nobody of us could stand at the site for the whole period), the only method getting matchable values between field monitoring and controlled experiments is to express it like rainfall in millimeters water column. Field

monitoring of the biocrusts CO2 gas exchange was recorded using a semi- automatic cuvette system (ACS) as described in detail by Lange (2002). Full technical details of the ACS (Walz Company, Effeltrich, Germany) are given in Lange et al. (1997). We therefore focus on some major topics of the procedure here. The whole device is composed of two major parts, first the cuvette system itself that is exposed in the natural environment of the biocrust (Fig. 1c) and secondly the controlling and data acquisition unit together with two infrared gas analysers (IRGA) for CO2 ambient and CO2 samples (Binos, Rosemount, Hanau, Germany) and a pumping unit regulated by mass flow controllers (Fig. 1e). For safety reasons a data printer and a graphics plotter were added as well. The soil crust samples were exposed on the lower part of the cuvette (Fig. 1d, arrow). When the upper lid was open (H in Fig. 1d), the sample was fully exposed to the natural environment. Measurements were taken every 30 minutes during which the cuvette was closed for 3 min. We recorded the CO2 exchange of the sample and absolute ambient CO2 partial pressure as well as mass flow, air temperature, the sample surface temperature, air humidity, and ambient photosynthetic radiation at the samples level. Net photosynthesis and DR were related to the area covered by the biocrust".

However, Fig 3 suggests (showing the time overlaps among bars of different colours) that sometimes through the year only one sample was measured, whereas in others, two, three, four or even five samples were measured. Were there any replicates at certain times of the year? If this is so, I think that this diversity in number of replicates along the year requires some comment. If this is not the case, the Fig 3 should be corrected or explained. Independently from the number of replicates during the monitoring period, the use of several different samples throughout the year would have been probably necessary, because the cyanobacterial biocrust samples have a limited resistance to handling and, after a series of measurements, they should be replaced. (Due to this fact, this is not properly a case of repeated measures over time). But again Fig 3 shows how the duration of the different samples is very different; in some cases the same sample appears to have been used repeatedly during even five or six

weeks, but in others, only once. Is this related with the difficult to understand the last sentence of section 2.2 (page 4, lines 20 - 21)? Please, explain better how, "for the one-year monitoring", you used 21 samples (page 4, line 16) and, though from those 21 used samples, only 11 were selected "for the long-term monitoring" (page 4, lines 20-21). Do you mean that those selected 11 samples were used repeatedly over time while the other 10 were used only once? Why?. By the way, the caption of the Fig 3 should be completed; a caption must be self-explanatory. I think we can assume that replacing the samples along the year for the monitoring is acceptable, apart from possibly necessary, since the monitoring was made only on carefully selected samples of the brown cyanobacterial community dominated by Symplocastrum purpurascens, and each sample includes probably billions of cyanobacterial individuals, being a good representation of the whole community. Besides, the microbiota at a certain sampling point could change enough along the year, which decreases the importance of always sampling at the same point. On the other hand, it would be advisable to state explicitly the number of times in which measurements were taken during the monitoring (and when), avoiding the reader having to speculate or discover this from the figures. For example, writing, "twenty-five measurements were taken between November and April, once per week, on the dates shown in the Fig 3".

Answer: We omitted figure 3 as it was confusing. We replaced it with table 1 that says what sample has been used for what time period. We took great care that all samples hat 1) the same surface area and more or less the same surface community of cyanobacteria (excluding lichens and bryophytes) and all showed a comparable range (plus/minus 5%) of NP and DR rates. Of course, we could not guarantee that the microbiota were comparable, but at least they respired at more or less the same range.

The sentence (page 4, lines 24-25) "The response of NP and DR to WC was determined for light, temperature and WC" could be better written, to avoid the expression 'the response to WC was determined for WC'. I am not sure I have understood this paragraph, particularly after seeing Fig 4. According lines 30-31 of page 4, the

temperature-related NP and DR were determined by varying temperatures while keeping constant both light and WC; whereas the WC-related NP and DR were determined (lines 1-2 of page 5) at constant light and different temperatures (in addition to different WC, it is supposed).

Answer: We rephrased the sentence as it was indeed confusing (see content of metarial and methos above)

However, according the Fig 4, it seems that temperatures and WC were not crossed. On the other hand, the first step of the procedure was determining the effect on NP and DR of light in every level of WC, for constant optimal temperature. I wonder whether the determination of the effect on NP and DR of temperature by itself (for constant optimum conditions of light and of WC) was the procedure for establish that optimal temperature. If so, then this should be explicit and constitute the firs step. If not, why is studying the effect of temperature by itself important since the effect on NP and DR of WC was determined for every temperature (keeping constant light)? I do not think these experiments were badly done, only that this paragraph is difficult to understand and raises doubts. Since to test the effect of each of these independent variables, at least one of them remained constant, the design is not fully factorial. Probably the triple interaction is significant and, in such a case it would be interesting to understand the biocrust functioning under natural conditions, to study the NP and DR response to that triple interaction, rather than the responses to every independent variables more or less separately. Nevertheless, meanwhile, this work provides very valuable information.

Answer: we indeed measured the WC response curve at 22, 27, 32, 37, 42, and 47 °C. But show only the WC curve at 47°C. For all curves shown in figure 4, always to variables were kept constant. For the WC we kept light and temperature constant, for the light curve temperature was kept constant while WC was kept in a narrow range (e.g. 0.77 to 1.10 mm for the optimal NP, and for the Temperature curve we kept water content in a narrow range and light constant. In figure 8 (new figure 7 in the revised manuscript version, see attachment), we plotted all NP-measuring points of biocrust

from one year against light and air humidity as an integrating variable of temperature and precipitation and got a nice pattern where one can see how many wet up and dry down cycles the biocrust underwent during the year.

In page 4, line 31, is 1500 mol photons m2 s-1 a saturating light? And, how is "optimal WC" defined? A (very short) definition appears only much after, in the line 15 of page 6. On the other hand, is the whole procedure described in the last sentence of this paragraph (lines 2-4 of page 5) repeated for every temperature? This is almost obvious but I think that to say it explicitly would be better.

Answer: optimal WC is that water content of the biocrust or of an organism where it reaches 90% of its maximal $CO_2$ uptake rate. 1500 $\mu$mol photons m-2 s-1 is not fully saturating but we choose this value in order not to photoinhibit the cyanobacteria as the experiment lastet for several hour and under this circumstances the chance of photoinhibiotn is rather large. We rephrased that part of the methods for better understanding.

Results The lines in Fig 4 b are not attributed to any WC levels. A series of lines similar to those of graph from Fig 4a are expected here, if I understood adequately the methods. Caption of Fig 4 does not help to understand this; in the part referred to graph 4b, any reference to the WC levels is missing. If the graph from Fig 4b refers to the effect on NP and DR of the temperature while keeping constant both light and WC, a value of (optimal) WC is lacking in the graph. What are the lines of Fig 4c, has each sample one line of NP and one line of DR? How? Where are the six different temperatures, since in the graph 4b temperatures and WC are not crossed?. Why the graph 4c shows 47 C as constant? experimental temperature whereas, according the main text (page 5, lines 1-2), six temperature were crossed with different WC levels?.

Answer: The graph 4b indeed shows the reaction of NP and DR to an increased temperature while light and WC were kept constant (WC = 0.77-1.10 mm). This is now explained in the text and the figure legend.

Why was the graphed experiment made at 47 C instead of at the optimal temperature (32C)?. Perhaps the authors plotted a graph for every temperature and only show the last one; but, in such a case, what are the lines of Fig 4c?.

Answer: We measured the reaction of NP and DR to increasing WC at 22, 27, 32, 37, 42, and 47 °C but do show only the curve measured at 47 °C as the other curve show only have different NP ranges but expose a similar pattern, meaning that they always remain in a certain range of WC for optimal NP.

The wording of Methods and/or Results should be a bit improved.

Answer: we agree and tried to improve wording considerably

Conclusion: In page 10, line 29, the sentence "three months having a negative balance probably due to regrowth of the biocrust" is hard to understand since, by default, 'regrowth' implies growth, and growth requires net CO2 assimilation. Besides, I think that this sentence about the regrowth requires an explanation, defining what exactly means 'regrowth' in this case, since this is closely related with the hypothesis presented at the end of the introduction (page 3, lines 12-14). Perhaps the authors used here the word 'regrowth' to refer to the recovery of metabolic activity after the latent-life span of the dry season.

Answer: You are right, this is misleading. We used the words reestablishment (of the biocrust) and resurrection (of NP) insetad.

It would be also advisable a better definition of that hypothesis in the Introduction

Answer: yes, we did that in the Introduction too and avoided the word "regrowth" Technical corrections Page 4, line 2; the expression '(factorial design)' would be better than '(factorial analysis)'. Indeed, the experiment was factorial (although not fully factorial); but, no statistical analysis is explicit in the Method section.

Answer: we omitted this term and rephrased with "For the analysis of the effect of the different environmental factors (light-, temperature- and water content; termed environmental manipulations throughout the text) on net photosynthesis".

Page 4, line 19: 'NP'and 'DR' appear in that line, whereas they are defined after, in lines 24 and 25 of that page 4

Corrected

Page 5, line 27: "2" is lacking after 'Fig' Page 6, line 15: A dot or a semicolon seems advisable just before the last word of that line.

Corrected

Thank you very much for your effort and helping us to improve the manuscript

- -0.3
- -1.5
- 0
- 1.5
- 3
- 4.5
- 6
- 7.5
- 9
- 11.5  Net photosynthesis (µmol CO₂ m⁻² s⁻¹)

**Fig. 1.** New figure 7

---

## Author Comment (AC1) · 9 Nov 2017

As suggested by Referee 1, I used the term 'cyanobacterially-dominated' biocrust. Well, a number of colleagues, all native English Speakers (Great Britain, New Zealand, Australia, USA), suggested not to convert the noun 'cyanobacteria' to an adjective. As I am not a native English speaker but my coauthor is, I will accept her strong suggestion not to use 'cyanobacterially-dominated' and com back to the use of 'cyanobacteria-dominated'.

---

## Author Response (AR1)

**Response to the three referees**

**Referee 1 (M. Bowker)**

This paper tackles a challenging measurement problem: estimation of the net C flux of a biocrust community over a year in the field. Also, using a battery of controlled environment treatments, the authors determine the response of these biocrusts to 5 moisture, temperature, and light. Overall, the authors find that biocrusts are a net C-sink in this environment, but net production is only observed for a portion of the year. The strength of the paper is that the authors have amassed an impressive amount of data and are one of only a handful of groups to complete this type of estimate. The weaknesses are perhaps due to a weak expression of why it is so important to conduct this measurement, and why other similar measurements have been scarce, and an occasional propensity to dwell on details without clear explanation of why they are 0 important.

10 important. Below, I provide several suggestions to help revise the paper.

**Answer**: Thanks for the positive valuation. We tried hard to eliminate the weaknesses in expression and also to explain every detail and why they are important.

Major comments: 1. I understand that this study does not fit the typical hypothesis test framework, but nonetheless the authors could ensure that readers comprehend the more interesting elements of this work in the abstract, introduction and

- 15 throughout. We can be fairly confident that most persistent biocrusts have a positive C-balance, because if they did not they would cease to exist eventually. Readers may find it intriguing that despite this apparent tautology, it is difficult to actually observe net CO2 uptake in biocrusts. This is distressing given that due to their extent, biocrusts may be non-trivial players in the global C cycle today, and almost certainly were major players in early terrestrial communities. We need this information. The reasons are various, but 2 major ones are that the positive CO2 uptake only occurs during a small part of the year in
- 20 most studies, and it is difficult to separate C-balance of biocrusts from C-flux from organisms (microbes, roots) or minerals (carbonates) that occur below them. If the study is framed as outlined above, obtaining an annual measurement becomes much more intriguing to the casual reader and the importance of this endeavor is understood.

**Answer**: We rephrased the referring parts of the abstract, introduction and discussion and tried to make the aims and outcomes unambiguously clear throughout the text. We included every suggestion of the referee and hope we could clarify the unclear or weakly expressed parts.

2. Consider standardizing terminology for the one year monitoring (also called "monitoring of gas exchange") and the factorial experiment (also called "gas exchange under controlled conditions"). I might suggest "environmental manipulations" and "Field monitoring".

Accepted, we changed this in the manuscript.

3. Consider placing the material on P4 L8-13 in section 2.3, and P4 L14-21 in section 2.4. It might improve flow and understandability.

Accepted.

4. P 4 L9 - Why were the samples stored frozen? This does not seem like a region where freezing soils are natural. Aren't 5 you worried this exposure could have harmed or otherwise altered the samples?

Answer: we added the following information for better understanding. "....This treatment had been tested in our laboratory many times with lichens of many different geographical origins, including the tropics and resulted in high survival rates (roughly 95%) compared to dry storing in herbarium cabinets or boxes in the laboratory. Earlier gas exchange measurements on biocrusts, cyanobacteria, byrophytes and lichens before freezing and after thawing and remoistening resulted in identical rates (unpublished laboratory tests)".

10

5. I could benefit from a few more details about how the 21 samples were used. For example you say 9 were used for the environmental manipulations, and 11 were used for long-term monitoring. What about the 21st sample? Also, I understand you inserted different biocrust samples for different portions of the field monitoring. But why are the samples used for such wildly varying times, I would have thought each would be used about 1.1 months?

- 15 Answer: We understood that this graph was confusing, as well as the text. We replaced the figure by Table 1, where we give just the sample and from when to when it was used. We also explained the somehow chaotic seeming randomization of the sample changing mode using the following sentence: "Fourteen samples were used during field monitoring (Table 1) and exposed in a random mode. The "random" mode was determined by the ability of access (climatic conditions, days off) by one of us to the investigation site during the whole measuring period".
- 20 6. There are 10 figures, are they are really needed? The content of Figure 6 is mentioned by the authors several times, but it is not completely clear to me why the authors ascribe so much importance to these 3 days. Also, figures 9 and 10 could probably be combined into one 2 panelled figure.

Answer: We omitted figure 3 and replaced it by a much clearer table. Figures 9 and 10 were combined as suggested so that the number of figures is now reduced to 8.

7. There are times when I would like to see different pieces of information integrated, and another case where there is 25 integration but I do not have all the information I need to understand it. Fig. 4. Provides plots of biocrust responses to different environmental gradients in a manner often used by this author group and associates. This is fine, but what I haven't ever seen is a plot integrating more than one of these variables in 3 dimensions. This would be a nice addition, if it could be done. Fig. 8 is a valiant attempt at illustrating responses to 2 environmental variables as a surface, but there is no explanation

of how this was created (Kriging?); further, the plot contains many inexplicable peaks and valleys, often near each other. Does this suggest overfitting? Maybe more aggressive smoothing is warranted.

Answer: Figure 8, which is now figure 7 is a contour plot made with the SigmaPlot Software and is based on a linear interpolation between measuring points (the same as in line graphs). We changed the figure legend in order to make it more understandable for readers. Each and every data point of net photosynthesis (measurements at daylight) was related with the referring air humidity and amount of light at the time of measurement. The colour indicates  $CO_2$  uptake rates (positive) or  $CO_2$  loss rates during the day (negative values) or inactivity (0; yellow colour). Here the new legend of figure 8 (formerly 7): "*Contour plot of net photosynthesis of the Boodjamulla biocrust based on linear interpolation between measured values. Shown is the active period from November 2010 to April 2011. Net photosynthesis is related to relative air humidity and photosynthetic active radiation (PAR). No dark respiration values shown! Colour key: yellow = no activity, orange to red = CO\_2 loss during the day (suprasaturation), light green to violet = CO\_2 uptake". We find this type of presentation impressive as it really shows how many wet up and dry down cycles a biocrust experiences in its natural environment and how common suprasaturation is.*

- 15 8. The discussion is not bad as written. You do address a key measurement issue, and hypothesize that the isolation of biocrust samples from underlying soil is the reason some studies find net C-uptake, and some find net C-loss. I would have like to see you more fully develop a few other elements too (several of which you do address to some degree), for example the generality that biocrusts maintain their existence by attaining positive C-balance only during a portion of the year, and that often the gains over a year are marginal. This means that oft-cited slow natural growth rates likely are due to
- 20 environmental constraints; only a minority of the year is actually suitable for growth. I would have like to see you advance some hypotheses for why different regions have different annual C-flux values. Related to this, one novel aspect of your study is that all other annual flux measurements were conducted in environments with cool season hydration. Finally, you could develop more your hypothesis about how expected climate changes might impact these naturally occurring biocrusts. It might be helpful to break the discussion into a few subsections devoted to distinct discussion topics.
- 25 Answer: The discussion is more or less newly written and also separated into several subsections. Here the new discussion:
  - 4. Discussion

[revised manuscript text omitted]

Minor comments Throughout: I suggest using "cyanobacterially dominated" (adverb modifying adjective) or "cyanobacteriadominated" (noun functioning to modify adjective), not "cyanobacteria dominated"(no hyphen, no adverb)

10 As suggested by Referee 1, I used the term 'cyanobacterially-dominated' biocrust in the revised version. However, a number of colleagues, all native English Speakers (Great Britain, New Zealand, Australia, USA), suggested not to convert the noun 'cyanobacteria' to an adjective. As I am not a native English speaker but my coauthor is, I will accept her strong suggestion not to use 'cyanobacterially-dominated' and came back to the use of 'cyanobacteriadominated.

15 P1L18 - remove "at"

Done

P1L19 - remove "during", suggest replacement of "referring" with "corresponding"

Done

Done

P2L21-23 - standardize terminology for net C-uptake, 3 different synonyms are used here

20

P2L27 – This would be a good place to mention that apparent C-source behavior is probably due to the challenges of properly measuring biocrust C-flux

Accepted and included

P4L15 - your meaning is unclear in the phrase "making sure that the area related..range"

25 Taken into regard and replaced by a new sentence: "All samples used were tested for a comparative large NP and DR rate under the given environmental conditions for two measurements (1 hour) in the cuvette system and only those were used that had more or less identical NP and DR rates".

P4L18 - suggest "drainholes" rather than "borings"

**Done**

P7L1 - suggest "monitoring" rather than "investigation"

Done

P7L29 - suggest "continuing" rather than "continued"

5 Done

P8L12 – that biocrusts are typically losing C does not mean that overall they are a C-source.

Accepted and expressed accordingly

P8L15 - Omit "When"

Done

10 Thank you Matthew for your very helpful comments.

**Referee 2 (anonymous)**

15

GENERAL COMMENT: This is a very nice study providing some data on a topic for which data is quite scarce: a hightemporal-resolution examination of isolated biocrust C exchange on an annual time scale in natural conditions. My main recommendation is to work on the framing of the study and improving the context of the results. These could be more

- compelling in a couple ways. First, the setting up at the beginning of a "mystery" that some crusts take up C and others don't does not work for the structure of this paper. I already knew the answer before seeing the results that it was a difference in methods among the studies (isolated crusts vs. whole soil column) so the results did not resolve this part of the narrative.
- Instead of taking this 'source vs. sink' approach to framing the study, I recommend more directly addressing the contribution of isololated crusts vs. other components of the ecosystem, which though not measured here, are measured much more often than isolated crusts in other studies. This brings me to the context of the results. The order of magnitude of one of the main results here, the 1.72 g m-2 is quite interesting (a very significant finding for which it is hard to come by data). It would be worth putting this in the context of eddy flux tower-measured NEE values, which tend to be on the order of 10s to 100s of g C m-2. For example, a recent study of drylands showed NEP varying from 350 to +330 g C m-2 (Biederman et al. 2017
- 25 http://dx.doi.org/10.1111/gcb.13686).

Answer: Yes, you are right and we agreed to take a somehow other framing. With the comments of the two other referees, we now used two clearly addressed aims (ass suggested by you): 1) how do the cyanobacterially

dominated biocrusts of Boodjamulla respond to the pronounced seasonality of water availability? and 2) are they sources or sinks for carbon at an annual timescale? We also discussed our results in relation to the outcome of the Biederman et al study: "In a recent study using the Eddy covariance method, Biederman et al. (2017) found a wide range of carbon sink/source function with mean annual net ecosystem productivity (NEP) varying from -350 to +330 g C m-2 across sites with diverse vegetation types in the dryland ecosystems of southwestern North America using evapotranspiration (ET) as a proxy for annual ecosystem water availability. Gross ecosystem productivity (GEP) and ecosystem respiration (Reco) were negatively related to temperature, both interannually within sites and spatially across sites and sites demonstrated a coherent response of GEP and NEP to anomalies in annual ET. Their investigation sites included one region having a noteworthy biocrust cover not accompanied by a dense vascular plant vegetation, the La Paz region of Baja California with an annual C-uptake (NEP) of roughly 90 g m-2. Approximating annual C gain based on the maximal CO2 uptake rates of four biocrust types composed of cyanobacteria, cyanolichens and chlorolichens measured by Büdel et al. (2013) from Baja California, we approach an annual C gain of those biocrusts of  $11 \pm 4$  g m-2 (calculation based on 90 active days per year with 34 of them having a sub-optimal CO2 uptake rate of only 25% of maximum due to suprasaturation. Daily rates were calculated by maximum NP for 5 hours per day minus 10 hours R + DR). This is 6.5 times more than our pure cyanobacterially dominated biocrust from Boodjamulla but still 8 time less than found for the Baja California site in the study of Biederman at al. (2017). It could well be that later successional biocrusts with a wealth of different species groups, including bryophytes, lichens and green algae besides of cyanobacteria, might reach even higher annual carbon fixation rates".

20 Furthermore, the other cited studies that do not isolate crusts report values on the order of 50 g C m-2 losses, and these losses likely come from plant roots (which the authors should discuss).

Done

5

10

15

25

30

The results in this study suggest that net exchange from these biocrusts are 1-2 orders of magnitude below the flux magnitudes of plants, possibly suggesting they are not a huge part of the total ecosystem C budgets. The Elbert et al review claims global NPP by cryptogamic covers to be 7% globally but that seems hard to reconcile with this type of result. Regardless of whether the authors totally agree with my reasoning here, this kind of context is worthy of more exploration in lieu of the source vs. sink issue.

Answer: We added the following paragraph: "Considering the Boodjamulla biocrust as a mid-successional type, where an increase in carbon gain might be expected in the future when lichens and bryophytes establish to form a later successional soil crust. Dealing here with a cyanobacterially dominated biocrust of a mid-successional type might explain the low C-uptake rates and also the seeming discrepancy to the values calculated for the global NPP by cryptogamic covers of Elbert et al. (2012)". The true strength of this study, the knowledge gap it fills, is in helping to understand the annual fluxes of an isolated crust into our overall understanding of ecosystem C exchange in systems that contain biocrusts. For that, it is a great study and certainly worthy of publication. The exploration of seasonality and environmental effects on flux rates are also quite nice and well done.

Thank you very much for this nice comment

**SPECIFIC COMMENTS**

5

15

20

abstract: "Of the metabolic active period, 48.6% were net photosynthesis and 51.4% dark respiration." 48.6% of what? measurement timepoints?

Answer: we rephrased that sentence for more clarity

10 typo: "above a relative humidity above 42%"

**Done**

Abstract: "This must be taken into consideration for future analyses and modelling of carbon balances in comparable biocrust ecosystems." I think a much more solid conclusion can be given here. It's not clear what comparable biocrust systems are being referred to and "it might be helpful for modeling" is never very compelling. I would end with direct and clear answers to the questions and what the results mean to the larger study of biocrust C exchange.

Answer: You are right we omitted this sentence and tried to give a more meaningful outlook in the conclusions: "From the magnitude of values it is clear that the observed C fluxes are not at all close to what a plant community can do. Methodological approaches analysing the carbon cycling of biocrusts need to critically reflect, that including or excluding sub-biocrust partitions might influence the status of the biocrust as being either considered as a sink or a source. There is an urgent need for more long-term measurements on different biological soil crust types and developmental stages from all climatic regions of the world".

Take a look at the significant digits throughout the manuscript. eg 31.66 nmol feels unrealistically and unnecessarily precise. This will make it easier to read too.

Done

25 P6 L5: missing close parenthesis?

Done

I'm not wild about the reporting of WC as mm precip equivalent. Wouldn't moisture levels depend on both precip in as well as evapotransipiration and drainage out? Why not report as something intuitive such as g water g-1 wet soil (ie % water by mass)? If kept, include an explanation of why this approach was used.

Answer: We added the following sentences in Material and Methods to explain why we use mm water column instead of g water per g biocrust: "Water content of samples of the experimental manipulations and those of the field monitoring is always expressed as millimeter water column. As it was impossible to remove the sample after each measurement from the monitoring cuvette system to determine the fresh weight corresponding with the measured value by weighing it with a balance (measurements every half an hour, nobody of us could stand at the site for the whole period), the only method getting matchable values between field monitoring and controlled experiments is to express it like rainfall in millimeters water column".

I like figures 1 and 2. I wish more papers would include nice methods photos like this.

Thanks for the nice comment

5

15

10 Figure 3 caption needs to define what the codes on the horizontal axis (x axis?) mean. I think it should say x axis and it says sample signature, which I didn't understand.

Does that mean an individually collected sample? I counted them and see 21 so I am guessing that's what it means. The codes are a bit cryptic.

Answer: we omitted figure 3 as it was by far too unclear. All three referees had problems with it and we no replaced by the much easier to understand table 1.

P2 L 23. what is meant by "carbon deposition"

Answer: should be C-uptake, we changed that in the sentence.

P2. L25. Part of the issue here is that studies are being mixed in which the biocrusts are by themselves or sitting on top of intact soil in which other fluxes like sub-crust respiration and root respiration contribute. This study uniquely looks at an

20 isolated crust with very high temporal resolution. It is quite interesting that net uptake is on the order of 1g m-2 year. This is quite meaningful when comparing with eddy flux values from drylands and many ecosytems that show NEE values on the order of 10s or 100s of g C m-2 year. This suggests that cyano crusts alone are not huge contributors to C uptake, even in dryland ecosystems. (Note: I jotted this paragraph down before reading the results and discussion)

I'd phrase the crucial question part of the intro more clearly and flag them as such with numbers and questions marks. Something like: This led us to two questions: (1) How do Boodjamulla biocrusts respond to the pronounced seasonality of water availability? and (2) Are they sources or sinks for carbon at the annual timescale?

Answer: thank you for this really good suggestion that we accepted and included.

"So far, it is unclear what is triggering a biocrust as either a sink or a source of C." I'm not sure this is conceptually accurate. If the biocrust itself, isolated as you have done here" is a persistent source, it dies. If it's a sink, it is growing or at least building SOM. I think what's unclear is the role of the crusts in the larger ecosystem and THAT'S what this study shows probably better than any previous study.

Answer: we agree and changed this part too.

P8 L10. A key point here though is that that top soil chamber still includes plants because there are plant roots below those crusts. The same is true for all the chamber approaches, and is one of the key reasons studies of isolated crusts like this one is valuable and needed. Those numbers (48.8 and 62) are likely large because they are catching the flip side of plant photosynthesis via roor respiration, which is a lot higher than what the crusts are doing. Chamber depth is more or less irrelevant because they are not closed on the bottom. Any root or microbe from anywhere below the measurement area would contribute to surface fluxes.

10 Answer: we agree and tried to express this in the Discussion.

In conclusion, some key take-homes from this study I'd like to see emphasized more are that the cyano crust alone is capable of being a small but consistent sink of carbon as it grows and possibly contributes a bit to SOM. However, it's clear also from the magnitude of the values that we are not seeing C fluxes that are anywhere close to what the plant community can do. A more complete discussion of these quantities would greatly improve the presentation of this excellent data set.

15 Answer: Again, we fully agree and tried to express that accordingly.

We thank the referee for heling us to improve the manuscript!

**Referee 3 (anonymous)**

GENERAL COMMENTS This work deals with the metabolic activity and gas exchange of a cyanobacterial biocrust in the Boodjamulla National Park of Australia. The authors have identified the main species in the biocrust, have carried out lab measurements of net photosynthesis (NP) and dark respiration (DR) as related with variable conditions of light, temperature and water content (WC), and have carried out one-year monitoring of NP and DR under field conditions, recording also a series of micro-climatic data.

The work is very interesting, in the cutting edge of knowledge, fully matches with the scope of the BG special issue, and is well made and well written with almost the only exception of some aspects in the methods section, which has consequences in part of the results. However, I think all my comments can be resolved with a few changes.

The results include the time duration of metabolic activity of the biocrust, the time proportion of NP vs DR, and an explanation of the timing of these processes. It should be noted that the authors provide with annual net CO2 flow data. They also provide evidence of strong seasonality of these biocrust-atmosphere gas exchange processes, and of the positive annual

balance of NP despite the relatively short time during which NP is achieved, showing that the net incoming CO2 flux produced by NP is larger than the flux due to DR, since the duration of NP seems shorter. The positive annual balance of CO2 is important because it is not obvious, having into account the low frequency of the conditions in which positive NP can be observed in situ in biocrusts in most of the sites.

5 Discussion and Conclusion are based in the results, include substantial contributions to knowledge and, along with the Introduction, show the experience and up-to-date knowledge of the authors.

Thank you for the positive valuation.

**SPECIFIC COMMENTS**

10 Methods. The sampling is a bit confusing. How many samples/replicates were used for the experiment of CO2 exchange under controlled conditions in lab? It seems that three replicates were used but the sentence in lines 12 and 13 of page 4 introduces some doubts. It is unclear what '9 different samples' are. If I have understood well the main text, I suggest rewriting that sentence, for example: "For every independent variable (light, temperature and water content), a different set of three samples/replicates was used". About the samples used for the one-year monitoring, taking some replicates in each 15 measurement time would have been better.

I think that it would been possible with only one cuvette (measurements in a series of replicates can be done in a short enough period to avoid that daily gas exchange variation had a significant effect in part of the replicates with regard to the others).

Answer: you are right, that was confusing and we rephrased the whole part as follows:

2.3 Environmental manipulations

For the analysis of the effect of the different environmental factors (light-, temperature- and water content; termed environmental manipulations throughout the text) on net photosynthesis, samples were air dried in a 10 cm Petridish, sealed and transported to the laboratory, where they were stored frozen (-20°C) until used for the measurements. This treatment had been tested in our laboratory many times with lichens of many different geographical origins, including the tropics and resulted in high survival rates (roughly 95%) compared to dry storing in herbarium cabinets or boxes in the laboratory. Earlier gas exchange measurements on biocrusts, cyanobacteria, byrophytes and lichens before freezing and after thawing and re-moistening resulted in identical rates (unpublished laboratory tests). Prior to the measurements, samples were allowed to defrost at 23°C for 12 h in an air tight box at low light intensities (<< 50  $\mu$ mol photons m-2 s-1) in order to avoid decondensation. Subsequently samples passively dehydrated and were kept at 23 °C and natural day-night cycles (~150  $\mu$ mol photons m-2 s-1) for 2

25

30

days. Light-, temperature- and water content related NP-measurements were performed using three independent replicates each. CO2 gas exchange measurements were conducted under controlled laboratory conditions using minicuvette systems (CMS 400 and GFS 3000, Walz Company, Effeltrich, Germany). The response of net photosynthesis (NP) and dark respiration (DR) was determined independently for light, temperature and water content (WC). Samples were weighed between measurements and WC was calculated later on as mm precipitation equivalent after final determination of the samples dry weight (exposed 5 days in a desiccator over silica gel at the end of the measurement). To obtain the NP response to light, fully hydrated samples (n=3) were exposed to stepwise increasing photosynthetic active radiation (PAR) from 0 to 2500 µmol photons m-2 s-1, near optimal temperature (32°C) and ambient CO2 concentration. The light cycle (about 30 min duration) was repeated until the samples were completely dry (after 3-4 h). Light saturation was defined as the PAR at 90% of maximum NP. The temperature related NP and DR were determined at increasing temperature steps, 22, 27, 32, 37, 42, and 47°C, while light was constantly at 1500 µmol photons m-2 s-1 and WC was constantly at optimum (n=3). The influence of WC on NP and DR was determined at constant, nearly saturating light (1500 µmol photons m-2 s-1) and six different temperatures (22, 27, 32, 37, 42 and 47°C) using three replicates. Samples were completely soaked with water and exposed in the cuvette. Then, NP and DR were measured in short time intervals (roughly 10 minutes) until the samples were almost dry and did not show any NP nor DR reactions. After each time interval, the fresh weight of the sample was determined using a balance and the corresponding WC to each data point calculated using the dry weight of the sample (see above).

In all experimental manipulations, the  $CO_2$  exchange rates of the sample were related to chlorophyll a content. For chlorophyll determination, the samples were ground to small pieces and then extracted two times with di-methyl-sulfoxide (DMSO) at 60 °C for 90 minutes. The chlorophyll a + b content was determined and calculated according to (Ronen and Galun, 1984).

**2.4 Field monitoring of CO2 gas exchange**

As there was only one semi-automatic cuvette system available, we could not replicate the measurements. To partly overcome this problem, we used several samples over the year. Samples were placed in a basket of thermoplastic resin with drainholes in the bottom to avoid standing water during rain events. The basket had a fixed size and all samples had exactly the same exposed surface of 16.5 cm2 (Fig. 1 d). All samples used were tested for a comparative large NP and DR rate under the given environmental conditions for two measurements (1 hour) in the cuvette system and only those were used that had more or less identical NP and DR rates. Fourteen samples were used during field monitoring (Table 1) and exposed in a random mode. The "random" mode was determined by the ability of access (climatic conditions, days off) by one of us to the investigation site during the whole measuring period.

10

20

15

30

Water content of samples of the experimental manipulations and those of the field monitoring is always expressed as millimeter water column. As it was impossible to remove the sample after each measurement from the monitoring cuvette system to determine the fresh weight corresponding with the measured value by weighing it with a balance (measurements every half an hour, nobody of us could stand at the site for the whole period), the only method getting matchable values between field monitoring and controlled experiments is to express it like rainfall in millimeters water column.

10

15

5

Field monitoring of the biocrusts CO2 gas exchange was recorded using a semi- automatic cuvette system (ACS) as described in detail by Lange (2002). Full technical details of the ACS (Walz Company, Effeltrich, Germany) are given in Lange et al. (1997). We therefore focus on some major topics of the procedure here. The whole device is composed of two major parts, first the cuvette system itself that is exposed in the natural environment of the biocrust (Fig. 1c) and secondly the controlling and data acquisition unit together with two infrared gas analysers (IRGA) for CO2 ambient and CO2 samples (Binos, Rosemount, Hanau, Germany) and a pumping unit regulated by mass flow controllers (Fig. 1e). For safety reasons a data printer and a graphics plotter were added as well.

- The soil crust samples were exposed on the lower part of the cuvette (Fig. 1d, arrow). When the upper lid was open (H in Fig. 1d), the sample was fully exposed to the natural environment. Measurements were taken every 30 minutes during which the cuvette was closed for 3 min. We recorded the CO2 exchange of the sample and absolute ambient CO2 partial pressure as well as mass flow, air temperature, the sample surface temperature, air humidity, and ambient photosynthetic radiation at the samples level. Net photosynthesis and DR were related to the area covered by the biocrust".
- 20 However, Fig 3 suggests (showing the time overlaps among bars of different colours) that sometimes through the year only one sample was measured, whereas in others, two, three, four or even five samples were measured. Were there any replicates at certain times of the year? If this is so, I think that this diversity in number of replicates along the year requires some comment. If this is not the case, the Fig 3 should be corrected or explained. Independently from the number of replicates during the monitoring period, the use of several different samples throughout the year would have been probably necessary,
- 25 because the cyanobacterial biocrust samples have a limited resistance to handling and, after a series of measurements, they should be replaced. (Due to this fact, this is not properly a case of repeated measures over time). But again Fig 3 shows how the duration of the different samples is very different; in some cases the same sample appears to have been used repeatedly during even five or six weeks, but in others, only once. Is this related with the difficult to understand the last sentence of section 2.2 (page 4, lines 20 21)? Please, explain better how, "for the one-year monitoring", you used 21 samples (page 4, lines 20 21)?
- 30 line 16) and, though from those 21 used samples, only 11 were selected "for the long-term monitoring" (page 4, lines 20-21). Do you mean that those selected 11 samples were used repeatedly over time while the other 10 were used only once? Why?.

By the way, the caption of the Fig 3 should be completed; a caption must be self-explanatory. I think we can assume that replacing the samples along the year for the monitoring is acceptable, apart from possibly necessary, since the monitoring was made only on carefully selected samples of the brown cyanobacterial community dominated by *Symplocastrum purpurascens*, and each sample includes probably billions of cyanobacterial individuals, being a good representation of the whole community. Besides, the microbiota at a certain sampling point could change enough along the year, which decreases the importance of always sampling at the same point. On the other hand, it would be advisable to state explicitly the number of times in which measurements were taken during the monitoring (and when), avoiding the reader having to speculate or discover this from the figures. For example, writing, "twenty-five measurements were taken between November and April, once per week, on the dates shown in the Fig 3".

5

10 Answer: We omitted figure 3 as it was confusing. We replaced it with table 1 that says what sample has been used for what time period. We took great care that all samples hat 1) the same surface area and more or less the same surface community of cyanobacteria (excluding lichens and bryophytes) and all showed a comparable range (plus/minus 5%) of NP and DR rates. Of course, we could not guarantee that the microbiota were comparable, but at least they respired at more or less the same range.

15 The sentence (page 4, lines 24-25) "The response of NP and DR to WC was determined for light, temperature and WC" could be better written, to avoid the expression 'the response to WC was determined for WC'. I am not sure I have understood this paragraph, particularly after seeing Fig 4. According lines 30-31 of page 4, the temperature-related NP and DR were determined by varying temperatures while keeping constant both light and WC; whereas the WC-related NP and DR were determined (lines 1-2 of page 5) at constant light and different temperatures (in addition to different WC, it is supposed).

Answer: We rephrased the sentence as it was indeed confusing (see content of metarial and methos above)

However, according the Fig 4, it seems that temperatures and WC were not crossed. On the other hand, the first step of the procedure was determining the effect on NP and DR of light in every level of WC, for constant optimal temperature. I wonder whether the determination of the effect on NP and DR of temperature by itself (for constant optimum conditions of

- 25 light and of WC) was the procedure for establish that optimal temperature. If so, then this should be explicit and constitute the first step. If not, why is studying the effect of temperature by itself important since the effect on NP and DR of WC was determined for every temperature (keeping constant light)? I do not think these experiments were badly done, only that this paragraph is difficult to understand and raises doubts. Since to test the effect of each of these independent variables, at least one of them remained constant, the design is not fully factorial. Probably the triple interaction is significant and, in such a
- 30 case it would be interesting to understand the biocrust functioning under natural conditions, to study the NP and DR response to that triple interaction, rather than the responses to every independent variables more or less separately. Nevertheless, meanwhile, this work provides very valuable information.

Answer: we indeed measured the WC response curve at 22, 27, 32, 37, 42, and 47 °C. But show only the WC curve at 47°C. For all curves shown in figure 4, always to variables were kept constant. For the WC we kept light and temperature constant, for the light curve temperature was kept constant while WC was kept in a narrow range (e.g. 0.77 to 1.10 mm for the optimal NP, and for the Temperature curve we kept water content in a narrow range and light constant. In figure 8 (new figure 7 in the revised manuscript version, see attachment), we plotted all NP-measuring points of biocrust from one year against light and air humidity as an integrating variable of temperature and precipitation and got a nice pattern where one can see how many wet up and dry down cycles the biocrust underwent during the year.

In page 4, line 31, is 1500 mol photons m2 s-1 a saturating light? And, how is "optimal WC" defined? A (very short) definition appears only much after, in the line 15 of page 6. On the other hand, is the whole procedure described in the last sentence of this paragraph (lines 2-4 of page 5) repeated for every temperature? This is almost obvious but I think that to say it explicitly would be better.

Answer: optimal WC is that water content of the biocrust or of an organism where it reaches 90% of its maximal  $CO_2$  uptake rate. This is also expressed now in the text. We rephrased that part of the methods for better understanding.

Results. The lines in Fig 4 b are not attributed to any WC levels. A series of lines similar to those of graph from Fig 4a are expected here, if I understood adequately the methods. Caption of Fig 4 does not help to understand this; in the part referred to graph 4b, any reference to the WC levels is missing. If the graph from Fig 4b refers to the effect on NP and DR of the temperature while keeping constant both light and WC, a value of (optimal) WC is lacking in the graph. What are the lines of

20 Fig 4c, has each sample one line of NP and one line of DR? How? Where are the six different temperatures, since in the graph 4b temperatures and WC are not crossed? Why the graph 4c shows 47 C as constant? experimental temperature whereas, according the main text (page 5, lines 1-2), six temperature were crossed with different WC levels?

Answer: The graph 4b indeed shows the reaction of NP and DR to an increased temperature while light and WC were kept constant (WC = 0.77-1.10 mm). This is now explained in the text and the figure legend.

25 Why was the graphed experiment made at 47 °C instead of at the optimal temperature (32C)?. Perhaps the authors plotted a graph for every temperature and only show the last one; but, in such a case, what are the lines of Fig 4c?.

Answer: We measured the reaction of NP and DR to increasing WC at 22, 27, 32, 37, 42, and 47 °C but do show only the curve measured at 47 °C as the other curve show only have different NP ranges but expose a similar pattern, meaning that they always remain in a certain range of WC for optimal NP.

30 The wording of Methods and/or Results should be a bit improved.

5

Answer: we agree and tried to improve wording considerably

Conclusion: In page 10, line 29, the sentence "three months having a negative balance probably due to regrowth of the biocrust" is hard to understand since, by default, 'regrowth' implies growth, and growth requires net CO2 assimilation. Besides, I think that this sentence about the regrowth requires an explanation, defining what exactly means 'regrowth' in this case, since this is closely related with the hypothesis presented at the end of the introduction (page 3, lines 12-14). Perhaps

5 case, since this is closely related with the hypothesis presented at the end of the introduction (page 3, lines 12-14). Perhaps the authors used here the word 'regrowth' to refer to the recovery of metabolic activity after the latent-life span of the dry season.

Answer: You are right, this is misleading. We used the words reestablishment (of the biocrust) and resurrection (of NP) insetad.

10 It would be also advisable a better definition of that hypothesis in the Introduction

Answer: yes, we did that in the Introduction too and avoided the word "regrowth"

Technical corrections Page 4, line 2; the expression '(factorial design)' would be better than '(factorial analysis)'. Indeed, the experiment was factorial (although not fully factorial); but, no statistical analysis is explicit in the Method section.

Answer: we omitted this term and rephrased with "For the analysis of the effect of the different environmental factors (light-, temperature- and water content; termed environmental manipulations throughout the text) on net photosynthesis".

Page 4, line 19: 'NP'and 'DR' appear in that line, whereas they are defined after, in lines 24 and 25 of that

page 4

15

Corrected

20 Page 5, line 27: "2" is lacking after 'Fig' Page 6, line 15: A dot or a semicolon seems advisable just before the last word of that line.

Corrected

Thank you very much for your effort and helping us to improve the manuscript

**Annual net primary productivity of a <del>cyanobacterially</del> cyanobacteria-dominated biological soil crust in the Gulf savannah, Queensland, Australia**

5 Burkhard Büdel1, Wendy J. Williams2, Hans Reichenberger1

4<del>Pland</del>-1Plant Ecology and Systematics, University of Kaiserslautern, Kaiserslautern, D-67663, Germany 2 Arid Soil Ecosystems, Agriculture and Food Sciences, University of Queensland, Gatton, 4343, Australia

**10 Correspondence to: Burkhard Büdel (buedel@bio.uni-kl.de)**

Abstract. Biological soil crusts (biocrusts) are a common element of the Queensland (Australia) dry savannah ecosystem and are composed of cyanobacteria, algae, lichens, bryophytes, fungi and heterotrophic bacteria. Here we report how the CO2 gas-exchange of the cyanobacteria-dominated biocrust type form Boodjamulla National Park in the north Queensland Gulf savannah responds to the pronounced climatic seasonality on the annual net primary productivity of a cyanobacterially and their quality as a carbon sink dominated biological soil crust form the Boodjamulla National Park in north western 15 Queensland using a semi-automatic cuvette system. The dominating dominant cyanobacteria are the filamentous species Sypmplocastrum purpurascens together with Scytonema sp. The recording period lasted from July 1st 2010 to June 30th 2011. Metabolic activity was found recorded between from November Juliy 1st 2010 until mid April June 30th 2011 only, referring to where CO2 exchange was only evident from November 2010 until mid April 2011, representative of 23.6% of the total time of the year. In November at With the onset of the raining-wet season in November, the first month (November) and the 20 last month (April) of activity had a pronounced respiratory loss of CO2. Also the last month of the raining season had a negative CO2-balance. Of tThe metabolic active period was accounted for 25% of the total time of the year wet season and of that 25% period, 48.6% were net photosynthesis (NP) and 51.4% dark respiration (DR). During the time of NP, Net-net photosynthetic uptake of CO2 during daylight hours was reduced by at 32.6% of the time of NP by due to water suprasaturation-during. In total, the biological soil crust fixed 229.09 mmol CO2 m-2 yr-1, referring-corresponding to an 25 annual carbon gain of 2.75 g m2 yr-1. Due to malfunction of the automatic cuvette system, data from September and October 2010, together with some days in November and December 2010, could not be analysed for net photosynthesis NP and dark respiration (DR). Based on climatic and gas exchange data from November 2010, an estimated loss of 88 mmol CO2 m2 was found for the two month, resulting in corrected annual rates of 143.08-1 mmol CO2 m-2 yr-1, equivalent to a carbon gain of 1.72 g m-2 yr-1. The bulk of the net photosynthetic activity occurred above a relative humidity <del>above of</del> 42%, indicating a 30 suitable climatic combination of temperature and, water availability, and a light intensity, well above 200  $\mu$ mol photons m-2  $s^{-1}$  photosynthetic active radiation. The Boodjamulla biocrust showed acxhibited highly seasonally varying varyability CO2 gas exchange pattern clearly divided into metabolically inactive winter month and active summer month. The metabolic

active period starts-commences with a period (of\_up to 3 month) of carbon loss, probably\_likely\_due to reestablishment of the crust structureregrowth and restoration of net photosynthesisNP effectivity\_before-prior to about a four\_-month period of net carbon gain. This must Seasonality iIn the Gulf savannah biocrust system, meant for-seasonality over the year investigated, showed that only a minority of the year is actually suitable for biocrust growth of the biocrust-thus a small window for end-a possible for potential contribution to soil organic matter (SOM) be taken into consideration for future analyses and modelling of carbon balances in comparable biocrust ecosystems.

**1. Introduction**

Biological soil crusts (named "biocrusts" throughout the text) are a consortium of heterotrophic bacteria, cyanobacteria, algae, fungi, lichens and bryophytes in different proportions with photoautotrophic organisms dominating their biomass.
They cover dryland soil surfaces and can make up to 70% of a dryland ecosystem's living cover (Belnap 1995; Belnap et al. 2016), but also occur in other climatic regions where competition with vascular plants is low (Büdel, 2001; Büdel et al., 2014). Due to the poikilohydric character of biocrust organisms, biocrusts exhibit a high resilience under extreme conditions and a remarkable adaptation to various combinations of climatic factors (e.g. Karsten et al., 2016; Sancho et al. 2016 and citations herein), thus making them excellent candidates for pioneering hostile environments on our planet. There is good

- evidence that eyanobacterial dominated cyanobacteria-dominated biocrusts have inhabited Earths soil surfaces at least 2600 million years ago (Watanabe et al., 2000; for an overview see also Beraldi-Campesi and Retallack, 2016). Lalonde and Konhauser (2015) point to the importance of oxygenic photosynthesis of early biocrusts providing sufficient equivalents for oxidative-weathering reactions in benthic and soil environments. This certainly also points to the role of biocrusts in soil formation and soil fertility—, for example by leaching carbon and nitrogen to initial soils. Consequently, there is growing interest in carbon gain of biocrusts (Lange and Belnap, 2016) and their CO2 exchange rates are considered relevant on local and global scales (e.g. Castillo-Monroy et al. 2011; Wilske et al. 2009; Elbert et al., 2012; Porada et al., 2013, 2014). Process based models as used by Porada et al. (2013; 2014) are-still based-rely on a few available datasets covering a small set number of BSC-biocrust types, organisms, geographical regions, and climatic situations (see also summary in Sancho et al.,

2016).

5

25 Regarding-With the focus on CO2 gas exchange of biocrusts on a long term basisover longer periods of time, a number of studies were published either on the basis of long term measurements or modelled from single or grouped measurements.
 From these results, Ttwo biocrust groups can be distinguished, one group-where biocrusts experienced carbon gain-net C-uptake and anthe other group-where biocrusts experienced carbon lossC-loss. Among Examples from the first-net C-uptake group\_include:7 a biocrust from the Mojave Desert exposed-gained a net input of-11.7 g C m-2 yr-1 (Brostoff et al., 2005), a biocrust of the northern Negev Desert, Israel had a net carbon deposition C-uptake of 0.7-5.1 g m-2 yr-1 (Wilske et al., 2008).

30 biocrust of the northern Negev Desert, Israel had a net carbon deposition C-uptake of 0.7-5.1 g m-2 yr-1 (Wilske et al., 2008, 2009), and a biocrust from a desert region of northwest China showed a carbon sequestration net C-uptake of 3.46-5 to 6.05

example, are a biocrusts of southeast Utah that was found to be a determined to be typical net carbon C-sources (Bowling et al., 2011), a biocrust in-from the Colorado Plateau, USA was-also losing  $62 \pm 8$  g C m-2 yr-1 (Darrouzet-Nardi et al., 2015), and a finally biocrusts from the Gurbantunggut Desert, Northwestern China showed exposed a C-release from of  $48.8 \pm 5.4$ to  $50.9 \pm 3.8$  g C m-2 yr-1 (Su et al., 2013). So far, it is unclear what is triggering a biocrust as either a sink or a source of Co One can be fairly confident that persistent biocrusts must have a positive C-balance. If they did not, they would certainly disappear from the reference habitats, what they in most cases do not. However Yet, despite the plausibility that biocrusts must have a net C-balance, it is difficult to observe net CO2 uptake. There are -various reasons however, -are various but-two major ones are that: 1) positive  $CO_2$  uptake might only occur during a small part of the year and 2) it is difficult to separate 10 the C-balance of the biocrust from C-fluxes of other organisms like microbes and roots of higher vegetation or minerals like carbonate that occur below them. As mentioned above, biocrusts played an important role in early terrestrial communities and might have been responsible for the first soils to develop by leaching carbon and nitrogen to initial soils. But why do we observe so heterogeneous results? One reason might be how these results were gained. As ButNevertheless, biocrusts are only one constituent of mature soils and, it seems plausible that measurements that include soil layers other than the biocrust 15 itself might result in CO2 release because of a high percentage of heterotrophic organisms (Bowling et al., 2011; Darrouzet-Nardi et al., 2015; Su et al., 2013), while those that restrict strictly to the biocrust layer might explain why they show are found acting as carbon sinks-net C-uptake over the year (Brostoff et al., 2005; Wilske et al., 2008, 2009; Feng et al., 2014). We believe that seasonality (biocrust wet-up and dry-down) plays an important role too for several reasons-too::. as-For example do cyanobacterial colonies exposed to wet-dry cycles apparently not fully recover, will with in that and quite a high percentage-number of cells just-die during the dry period (e.g. Grilli -CVaiola, et al., 1993; BilliGrilli Cailola and Billi, 20 2006). Another important finding observation considered in for the design of our the present study presented here iwas, that the determination of CO2 gas exchange of single species might not represent the biocrust. There is a strong influence on the outcome of the measurements when species are removed from the context of the biocrust-context, rather than studying the whole BSC biocrust system-(Colesie et al., 2016; Elbert et al., 2012)-.), as Tthis does not necessarily represent the ecological response of an intact biocrust (Weber et al., 2012). 25 Previously, it was observed Coming from the eobservation that we could not resurrect the Australian Gulf savannah

1 g C m-2 yr-1 (Feng et al., 2014). On the other hand, where Among the carbon losing biocrusts ecosystems lose carbon, for

biocrusts photosynthetic activity in the middle of the dry season, even after soaking them in water for more than 24 hours
 (Williams et al. 2014)7. This motivated us to perform a long term study on an entire lycyanobacteria-dominated biocrust
 common in northern Queensland, that can also be considered as a mid-successional type in a highly seasonal environment.
 Our main research-All theise considerations led us to the questionsquestion were: i1) hHow do the eyanobacterially
 dominated cyanobacteria-dominated biocrusts of Boodjamulla fast does a cyanobacteria-dominated biocrust resurrect positive
 net gas exchange after a long period of drought respond to the pronounced seasonality of water availability? and ii2) is there
 a seasonal behaviour in terms of carbon gain in the growing season apart from the annual wet dry cyclesAare theyse

biocrusts sources or sinks for carbon at an annual timescale? , and iii) we hypothesize that there must be considerable regrowth of the crust forming cyanobacteria before the will we find biocrust performs positive net photosynthesiscarbon gain over the period of one year if restrict measurements to the biocrust only?

**In this study**

- 5 Here we focused on a common biological soil crust types occurring in the Gulf Plains bioregion covering 8,868 km2 
[revised manuscript text omitted]
 (~150  $\mu$ mol photons  $m_2^{-2}$  s-1) for two days. All-measurements-Light-, temperature- and water content and related NPnet photosynthesis (NP) -measurements were done-performed with-using three independent samples replicates each-(3 treatments with 3 samples = 9 different samples). CO2 gas exchange measurements were conducted under controlled laboratory conditions using minicuvette systems (CMS 400 and GFS 3000, Walz Company, Effeltrich, Germany). The response of net
- 15 photosynthesis (NP) and dark respiration (DR) to water content (WC) was determined independently for light, temperature and water content (WC). Samples were weighed between measurements and WC was later-calculated later on-as mm precipitation equivalent after final determination of the samples dry weight following (exposed 5 days in a desiccator over silica gelat the end of the measurements). To obtain the NP response to light, fully hydrated samples (n=3) were exposed to stepwise increasing photosynthetic active radiation (PAR) from 0 to 2500 μmol photons m-2 s-1-, near optimal temperature
- 20 (32°C) and ambient CO2 concentration. The light cycle (about 30 min duration) was repeated until the samples were completely dry (after 3–4 h). Light saturation was defined as the PAR at 90% of maximum NP. The temperature related NP and DR were determined at increasing temperature steps, from-22, 27, 32, 37, 42, and -47°C, while light was -at-constantly at 1500 µmol photons m-2 s-1 and WC was constantly at optimum optimal WC-(n=3). The reaction of influence of WC on NP and DR to different biocrust water contents were was determined at constant, nearly saturating light (1500 µmol photons m-2
- s-1) and six different different temperatures (For testing the reaction of NP and DR to biocrust WC, completely wet samples (n = 3) were exposed in the cuvette system and measured at 6 different temperatures (22, 27, 32, 37, 42 and 47°C), again using three replicates, steps, from 22, 27, 32, 37, 42 to 47°C (n = 3). Samples were completely soaked with water were and exposed installed in the cuvette, and Then, NP and DR were measured in short time intervals (roughly 10 minutes) until the samples were almost dry and did not show any NP nor DR reactions. After each time interval, the fresh weight of the sample was determined using a balance and the corresponding WC to each data point calculated using the dry weight of the sample
- 30 was determined using a balance and the corresponding WC to each data point calculated using the dry weight of the sample (see above).

[revised manuscript text omitted]

**3.1 CO2 gas exchange under controlled conditions Environmental manipulations**

When exposed to stepwise increasing PAR intensities, the biocrust did not reach full saturation of NP at optimal water content  $(31.\underline{766} \pm 2.64 \text{ nmol CO}_2 \text{ mg}^{-1} \text{ chlorophyll } a \text{ s}^{-1} \text{ at a WC of } 0.70 \pm 0.08 \underline{1} \text{ mm and } 32^{\circ}\text{C}; n=3)$  even at 2500 µmol 15 photons  $m^2$  s-1. At a-WC below the optimal WC (that i.e. WC, where at least 90% of the maximum gas exchange rates are reached), a decline of NP (21.28-3  $\pm$  5.69-7 nmol CO2 mg-1 chlorophyll a s-1 at a WC of 0.54  $\pm$  0.08-1 mm was observed7. as This was also the case for WC well above optimal WC,  $20_{2,66-7} \pm 6_{7,2}24$  nmol CO2 mg-1 chlorophyll a s-1 at a WC of  $01.970 \pm 0.14$  mm or  $7_{2.5}6 \pm 3_{2.7}3$  nmol CO2 mg-1 chlorophyll *a* s-1 at a WC of  $1.32 \pm 1.56$  mm and  $2.263 \pm 0.22$  nmol CO2 mg-1 chlorophyll *a* s-1 at a WC of  $1.88-9 \pm 0.13$  mm (Fig. 3a).

20

5

Increasing air temperature from 22 to  $47^{\circ}$ C resulted in an increase of NP from  $19.8 \pm 1.44$  nmol CO2 mg-1 chlorophyll *a* s-1 to  $32.\frac{39}{4} \pm 4.47.5$  nmol CO2 mg-1 chlorophyll a s-1 (n = 3) without saturation. The increase of dark respirationDR was less expressed and ranged from -3.09-1 nmol CO2 mg-1 chlorophyll *a* s-1 at 22°C to  $-6.33 \pm 1.37-4$  nmol CO2 mg-1 chlorophyll  $a \text{ s}^{-1}$  at 47°C air temperature (n = 3; Fig. 3b).

[revised manuscript text omitted]

**4. Discussion**

10

**4. 4.1 Seasonality and CO2 balances**

We found a Apart from a clearly seasonal activity pattern of the eyanobacterially dominatedcyanobacteria-dominated biocrust from Boodjamulla National Park, Queensland biocrust, only a minority of the year was actually suitable for its growth during the one year round  $CO_2$  gas exchange field monitoring. Approximately exposing Apart from Apart From Apart from the period with no

- 15 measurable CO2 gas exchange from-lasted from July to mid-September 2010 and then from mid-April to end of June 2011. Metabolic activity was found in the summer months only, starting with September 23rd 2010 with-where the first rains events commenced, and continueding until April 18th 2011. Due to malfunction of the ACS, measurements from September and October and some days of November and December 2010 were not useable to calculate NP and DR. An estimation based on rainfall data from September and October, together with the referring-reference gas exchange values from November suggests a CO2 loss of roughly 88 mmol m-2. Net primary productivity was determined as 1.72 g C m-2 yr-1 (2.75 8 g C m-2 yr-1 without Sept.-Oct. correction). Our results state-showed that the Boodjamulla biocrust exposed exhibited a positive net C-uptake act as a carbon dioxide sink after one year field monitoring, and tThis result is in line with the findings of several other studies but differs from all of them in the fact that our study focused to an environment with hot wet-season hydration, whereas all of the other studies were conducted in environments with cool season hydration. For example, of-a
- 2005), 6.7 times higher than the eyanobacterially dominated cyanobacteria-dominated Boodjamulla biocrust7 a-Another biocrust dominated by cyanobacteria, algae, lichens and mosses biocrust from the Negev Desert, Israel with exposed resulted in a C gain of 0.7 to 5.1 g m-2 yr-1 (Wilske et al., 2008, 2009) and thus is pretty close to what we observed in our study, and which also corresponds with the results -or-from biocrusts composed of a-cyanobacteria, lichens and mosses biocrust of the
- 30 Mu Us Desert in China with a C gain of 3.46-5 to 6.05-1 g m-2 yr-1 (Feng et al., 2014).

| I  | HoweverOn the other hand, there are several studies that clearly demonstrate that biological soil crusts can also act as                            |                          |
|----|-----------------------------------------------------------------------------------------------------------------------------------------------------|--------------------------|
|    | loose C sources to the atmosphere. When studying a symposeterially dominated wanobacterial dominated biocrust of the                                |                          |
|    | arid grassland in southeast Utab USA applying the Eddy covariance method. Bowling et al. (2010) could not decide if this                            |                          |
| 1  | biocrust is was a sink or a source as there were some grasses involved in the plot and hence their root respiratory CO. loss                        |                          |
| 5  | influenced the CO. However, wWhen these authors applied a ton soil chamber for gas exchange measurements, they found                                |                          |
| 5  | the same biocrust a twoical C source (Bowling et al. 2011). However, But still, this does not necessarily mean that overall they                    | Formatient: Theigesteint |
|    | are a C source A supplied C source (Bowling et al. 2011). However, but suit, this does not necessarily mean that overlan they                       |                          |
|    | are a C-source. A cyanometerial, include diornal from the Gurbantungut Descri, China was reported as quite a                                        |                          |
| I  | targe C source with a loss of -48.8 $\pm$ 5.4 to -50.9 $\pm$ 5.8 g C m yr (Su, Y. G. et al., 2012, 2013) and a very similar biocrust                |                          |
| 10 | type of the arid grassiand of the Colorado Plateau, USA that exposed surprisingly similar values of $-62 \pm 8$ g C m yr                     |                          |
| 10 | (Darrouzet-Nardi et al., 2015). How can this astonishing and at first glance contradictory fact be explained? When                                  |                          |
|    | eComparing methodology and how measurements were taken, sheds some light on this phenomenon. All investigations,                                    |                          |
|    | including our own study, that that showed to biocrusts having a net $CO_2$ -uptake over the year acting as sinks-used $CO_2$                        | Formatiert: Tiefgestellt |
|    | gas exchange devices with a separate cuvette where the samples had to be removed from the biocrust (Brostoff et al., 2005;                          |                          |
|    | Feng et al., 2014) except the study of Wilske et al. (2008, 2009) that used a top soil chamber measuring the biocrust in situ.                      |                          |
| 15 | All other studies used top soil chambers where the biocrust is measured in situ (Bowling et al., 2011; Su et al., 2013;                             |                          |
|    | Darrouzet-Nardi et al., 2015). The main difference we could find found-in all of thiese studies was the thickness of the                            |                          |
|    | biocrust plus sub-crust (soil) layer used. While those studies revealing the biocrusts as C sources as CO2 loosers used collars                     | Formatiert: Tiefgestellt |
|    | penetrating 20 to 35 cm deep into the soil (Bowling et al., 2011; Su et al., 2013; Darrouzet-Nardi et al., 2015), the studies                       |                          |
|    | attributing biocrusts to C sinks as CO2 winners during the course of a one year course, either used either pieces of biocrusts                      |                          |
| 20 | from 1 to 5 cm thickness (this study, Brostoff et al., 2005; Feng et al., 2014), or a collar penetrating only 5.5 cm into the soil                  |                          |
|    | (Wilske et al., 2008, 2009). The metabolic activity of heterotrophic organisms as well as respiration of roots from nearby                          |                          |
|    | plants_from_of_deeper soil levels apparently influence the CO2 gas exchange measurements accordingly as was already                                 |                          |
|    | indicated in the investigation of Bowling et al. (2011) on the biocrust underlying soil biotic community. Yet, soils are not a                      |                          |
|    | perpetual motion machine in terms of carbon balance, they can only respire as much carbon as is introduced into the system.                         |                          |
| 25 | If carbon does not come from the autotrophic part of the soil system, it must be introduced from outside, either via litter                         |                          |
|    | transport, blown dust, animals, or with run-on water from the surrounding environment. In a recent study using the Eddy                             |                          |
|    | covariance method, Biederman et al. (2017) found a wide range of carbon sink/source functionwithThere was a mean                                    |                          |
|    | annual net ecosystem productivity (NEP) varying from -350 to +330 g C m 2 2 across sites with diverse vegetation types in the | Formatiert: Hochgestellt |
|    | dryland ecosystems of southwestern North America using evapotranspiration (ET) as a proxy for annual ecosystem water                                |                          |
| 30 | availability. Gross ecosystem productivity (GEP) and ecosystem respiration (R eco ) were negatively related to temperature,              |                          |
|    | both interannually within sites and spatially across sites and sites demonstrated a coherent response of GEP and NEP to                             |                          |
|    | anomalies in annual ET. Their investigation sites included one region having a noteworthy biocrust cover not accompanied                            |                          |
|    | by a dense vascular plant vegetation, the La Paz region of Baja California with an annual C-uptake (NEP) of roughly 90 g m                          | Formatiert: Hochgestellt |
|    |                                                                                                                                                     |                          |

2. Approximating annual C gain based on the maximal CO2 uptake rates of four biocrust types composed of cyanobacteria, cyanolichens and chlorolichens measured by Büdel et al. (2013) from Baja California (Büdel et al., 2013), we approach an annual C gain of those biocrusts of  $11 \pm 4$  g m22. (This calculation was based on the following estimations: 90 active days per year with 34 of them having a sub-optimal CO2 uptake rate of only 25% of maximum due to suprasaturation. Daily rates were calculated by maximum NP for 5 hours per day minus 10 hours R + DR). This is-estimation resulted in 6.5 times more that our pure evanobacterially dominated cyanobacteria-dominated biocrust from Boodjamulla but still 8 time less than found for the Baja California site in the study of Biederman at al. (2017). It could well be that later successional biocrusts

with a wealth of different species groups, including bryophytes, lichens and green algae besides of cyanobacteria might

reach even higher annual carbon fixation rates. This should be in the focus of further studies.

10

**4.2 Carbon dioxide uptake rates and biocrust type**

Maximum net CO2 uptake rates of the Boodjamulla biocrust (8.3  $\mu$ mol CO2 m-2 s-1) clearly exceeded those of a comparable evanobacterially dominated cyanobacteria-dominated biocrusts from the Negev Desert. Israel that reaching 15 reached maximal values of 1.12 µmol CO2 
[revised manuscript text omitted]
 could be expected indirectly, when warming it is related to influences rainfall amount and
  - regime. We-It could be speculated that less, but heavier rain events weould certainly effect the Boodjamulla biocrust by increasing suprasaturation periods resulting in lower or even no carbon gain probably also causing a pronounced reduction in coverage.
- 25

**5. Conclusion**

The Boodjamulla biocrust showed a highly seasonal photosynthesis-related metabolic activity divided into four major periods: 1) the a metabolically inactive winter monthtime; 2) the onset of the photosynthetic active period, starting with roughly three month of reestablishment, having a negative balanceshowing and limited C-loss-CO2-uptake due to heavy suprasaturation, and a hypothesized increased activity of heterotrophic organisms decomposing organic matter from old biocrusts, probably due to regrowth and increased activity of heterotrophic organisms feeding on dead organic matter of of the old biocrust; 3) a four-month period of carbon gainnet C-uptake; 
[revised manuscript text omitted]
 = CO2 uptake. The graph was created using all gas exchange measures (half hourly) of corresponding values of air humidty and light.

Figure 8: A) Mean diel activity of the Boodjamulla biocrust; black = inactive, light grey = photosynthetically active, dark grey = dark respiration, hatched = metabolic activity but due to technical failure of instrumentation, not clear if NP or DR. B) Monthly extent of water suprasaturated periods during the photosynthetic (NP) active time of the Boodjamulla biocrust. Black = periods of suprasaturation, light grey = periods of conducive water supply.

35

**Supplementary figures**

30

Figure S1-S6: Each month with metabolic activity is shown. One month is represented as a set of three graph pairs (except April where metabolic activity ceased by mid-month), each pair composed of an upper graph showing CO2 gas exchangegas exchange (green and

40 black curves) and PAR (brown curve) and a lower graph showing ambient CO2 concentration (black curve), air temperature (red curve), and relative air humidity (blue curve).

Figure S1: Diel carbon dioxide gas exchange in November 2010

Figure S3: Diel carbon dioxide gas exchange in January 2011
Figure S4: Diel carbon dioxide gas exchange in February 2011
Figure S5: Diel carbon dioxide gas exchange in March 2011

5 Figure S6: Diel carbon dioxide gas exchange in April 2011

| Table 1 . Samples used for momorning (only mose listed used during the active period). |
|-----------------------------------------------------------------------------------------------|
|-----------------------------------------------------------------------------------------------|

| 1. Sample A10 Sep. $23^{st}$ – Dec. $12^{th}$ 2010 ( | (80 days) |
|------------------------------------------------------|-----------|
|------------------------------------------------------|-----------|

- $\begin{array}{l} \underline{\text{Dec. } 13^{\text{th}} \text{Dec. } 12 2^{\text{th}} 0 (00 (\text{tays}))} \\ \underline{\text{Dec. } 13^{\text{th}} \text{Dec. } 22^{\text{th}} 2010 (\text{J} \text{ Jan. } 4^{\text{th}} \text{Jan. } 8^{\text{th}}, \text{Jan. } 12^{\text{th}} \text{Jan. } 14^{\text{th}} 2011 (18 \text{ days}) \\ \underline{\text{Dec. } 23^{\text{rd}} \text{Dec. } 29^{\text{th}} 2010 (7 \text{ days})} \\ \underline{\text{Dec. } 30^{\text{th}} \text{Dec. } 31^{\text{th}} 2010 (2 \text{ days})} \\ \underline{\text{Tays}} \\ \underline{\text{$ 2. Sample C5 3. Sample S1
- 5 4. Sample SC
- 5. Sample S4 Jan.  $1^{st}$  – Jan.  $3^{rd}$ , Jan.  $9^{th}$  – Jan.  $11^{th}$  2011 (6 days)
- 6. Sample S7 Jan. 15th – Jan. 17th 2011 (3 days)
- Jan.  $18^{th}$  Jan.  $25^{th}$  2011 (8 days) 7. Sample 2B
- 8. Sample BS1 Jan. 26th – Feb. 1st 2011 (6 days)
- Feb.  $2^{nd}$  Feb.  $13^{th}$  2011 (12 days) 10 9. Sample BS2
- Feb. 14th Mar. 13th 2011 (28 days) 10. Sample BS4
- $\frac{\text{Mar. 14}^{\text{th}} \text{Mar. 24}^{\text{th}} 2011 (11 \text{ days})}{\text{Mar. 25}^{\text{th}} \text{Apr. 4}^{\text{th}} 2011 (11 \text{ days})}$ 11. Sample BS7
- 12. Sample BS3
- Apr.  $5^{th}$  Apr.  $17^{th}$  2011 (11 days) Apr.  $18^{th}$  Apr.  $20^{th}$  2011 (13 days) Apr.  $18^{th}$  Apr.  $20^{th}$  2011 (3 days) 13. Sample C14
- 15 14. Sample C11B

Table 1: Monthly net primary productivity of the Boodjamulla biological soil crust (values in brackets are an estimation only, not based on measurements; see text for explanantion).

|                   | NPP (NP-DR)                                              |                                                |  |
|-------------------|-----------------------------------------------------------------|------------------------------------------------|--|
| Month             | $(\text{mmol CO}_2 \text{ m}^{-2} \text{ month}^{-1})$          | (g C m-2 month-1) |  |
| July              | 0                                                        | 0                                       |  |
| August            | 0                                                        | 0                                       |  |
| September 2010    | 0 (-2.0)                                                 | 0 (-0.02)                               |  |
| October 2010      | 0 (-86.0)                                                | 0 (-1.06)                               |  |
| November 2010     | -210.26                                                  | -2.53                                   |  |
| December 2010     | 110.20                                                   | 1.32                                    |  |
| January 2011      | 99.58                                                    | 1.20                                    |  |
| February 2011     | 80.99                                                    | 0.97                                    |  |
| March 2011        | 174.11                                                   | 2.09                                    |  |
| April 2011 | -25.54                                                   | -0.31                                   |  |
| May 2011   | 0                                                        | 0                                       |  |
| June 2011  | 0                                                        | 0                                       |  |
| Annual            | 229.09 mmol CO 2 m -2 yr -1    | $2.75 \text{ g C m}^{-2} \text{ yr}^{-1}$      |  |
|                   | (143,08 mmol CO 2 m -2 yr -1 ) | (1.72 g C m -2 yr -1 )   |  |

---

## Author Response (AR2)

Reviewer 1

No requirements for changes or improvement.

Reviewer 3

General comment

This work deals with the metabolic activity and CO2 gas exchange over time of a cyanobacterial biocrust in the Boodjamulla National Park of Australia. The manuscript has improved after suffer many and substantial modifications from its original version. The abstract and the Introduction are now longer and their wording is better. Methods, in addition to be longer and better written, include modifications in their structure, and the wording and structure of the Results have been modified accordingly. The Discussion has been rewritten, is now longer, including new comparisons, is structured, and includes a section with new content. Conclusion has been rewritten and improved also. In my opinion, the manuscript is very good and ready to be published in its present version except for two small details: the uncertainty about the days in which the CO2 monitoring in field was made and the name of 'Table 1' in the page 17 (that table is now 'Table 2')

**Answer**: corrected, see comment below

Specific comments

Methods

I agree with the present version except for the uncertainty on the days of the field CO2 monitoring (see below). From the questions I posed in my previous revision and the changes I proposed, those which have not been answered, or accepted, individually have become obsolete because the Methods section has been rewritten. The contents of the Methods section are better structured now in subsections.

Particularly, the text of the section 2.3 of the manuscript (before titled 'CO2 gas exchange under controlled conditions', and now titled 'Environmental manipulations') has been deeply rewritten and is now more fully explained and much clearer than before. The number of used samples, how prepared, as well as the procedure followed to disentangle the response of NP and DR to variations in light, temperature and water content can be understood now without doubts.

Similarly, the text of the section 2.4 has been lengthened and deeply rewritten too. Only one detail remains unclear: the days in which the one-year field monitoring of CO2 was carried out are not yet explicitly stated. A reader could interpret that measurements were taken every 30 minutes during every day of the year. Was this so or was the field CO2 exchange measured in 34 different days spread between September 2010 and April 2011, as the Table 1 (the table replacing the old figure 3 of the previous version, now removed) seems to suggest?.

**Answer**: The reviewer is right and I changed this in the table as well as in the text of the manuscript. It is now clear that the measuring period was from July 1$^{st}$ 2010 until June 30$^{th}$ 2011. During this period measurements were taken every 30 minutes.

Results

I agree with the present version. After the considerable improvement of the wording of Methods section and the modifications in the figure captions, my difficulties with the old figure 4 (now Figure 3) have disappeared.

Discussion

I agree with the present version. I congratulate the authors for the inclusion of new text in the Discussion, for example quoting very recent works and showing the important differences among the $CO_2$ flux as calculated from direct field measurements on biocrusts and that estimated from Eddy Correlation systems. But particularly, for adding a new section on "Reestablishment and resurrection after the dry season", which constitute an overwhelmingly good response to my previous question about what exactly 'regrowth' meant in the conclusions, and questioning whether such a word (now removed) was adequate (since, by default, 'regrowth' implies growth, which requires net $CO_2$ assimilation)

Conclusion

I agree with the present version, which is better

References, Tables and Figures

I agree with the present version

Minor remarks

After remove the old figure 3, replacing it by the new Table 1, the other table, on the monthly net primary productivity (old Table 1), remains with the name of 'Table 1' in the page 17, although in the main text it appears as Table 2 (line 25, page 6)

     **Answer**:Corrected, thanks for the hint!

[revised manuscript text omitted]

Figure 1

[Figure]

recent cyano-
bacterial
growth layer

empty EPS
layers of
previous
years (1-2)

oldest EPS layer

soil particles

Figure 2

[Figure]

Figure 3

[Figure]

Figure 4

[Figure]

Figure 5

[Figure]

Figure 6

[Figure]

Figure 7

[Figure]

A

[Figure]

B

Figure 8